# Lipid-mediated regulation of SKN-1/Nrf in response to germ cell absence

**Michael J Steinbaugh[1,2], Sri Devi Narasimhan[1,2], Stacey Robida-Stubbs[1,2], Lorenza E Moronetti Mazzeo[1,2], Jonathan M Dreyfuss[1,3], John M Hourihan[2†], Prashant Raghavan[1,2], Theresa N Operaña[1,2], Reza Esmaillie[1,2], T Keith Blackwell[1,2]***

[1]Research Division, Joslin Diabetes Center, Boston, United States; [2]Department of Genetics and Harvard Stem Cell Institute, Harvard Medical School, Boston, United States; [3]Department of Biomedical Engineering, Boston University, Boston, United States

**Abstract** In *Caenorhabditis elegans*, ablation of germline stem cells (GSCs) extends lifespan, but also increases fat accumulation and alters lipid metabolism, raising the intriguing question of how these effects might be related. Here, we show that a lack of GSCs results in a broad transcriptional reprogramming in which the conserved detoxification regulator SKN-1/Nrf increases stress resistance, proteasome activity, and longevity. SKN-1 also activates diverse lipid metabolism genes and reduces fat storage, thereby alleviating the increased fat accumulation caused by GSC absence. Surprisingly, SKN-1 is activated by signals from this fat, which appears to derive from unconsumed yolk that was produced for reproduction. We conclude that SKN-1 plays a direct role in maintaining lipid homeostasis in which it is activated by lipids. This SKN-1 function may explain the importance of mammalian Nrf proteins in fatty liver disease and suggest that particular endogenous or dietary lipids might promote health through SKN-1/Nrf.

*For correspondence: keith.
blackwell@joslin.harvard.edu

Present address: †Research
Division, Joslin Diabetes Center,
Boston, United States

Reviewing editor: Kang Shen,
Howard Hughes Medical
Institute, Stanford University,
United States

## Introduction

The nematode *Caenorhabditis elegans* has been invaluable for identifying mechanisms that slow aging and may prevent chronic disease (*Kenyon, 2010*). An intriguing finding that was first made in this organism is that when germline stem cells (GSCs) are ablated, mechanisms are activated in somatic tissues that protect against stress and increase lifespan (*Hsin and Kenyon, 1999*; *Kenyon, 2010*; *Antebi, 2013*; *Hansen et al., 2013*). GSC loss also increases lifespan in *Drosophila melanogaster* (*Flatt et al., 2008*), and castration has been associated with longevity in men (*Min et al., 2012*), suggesting that this relationship might be conserved. These beneficial effects of GSC removal may have evolved to maximize reproductive fitness under adversity (*Partridge et al., 2005*; *Kenyon, 2010*). This relationship provides paradigms for how tissue non-autonomous signals influence aging (*Kenyon, 2010*), and how a stem cell population communicates with the 'niche' that sustains it (*Jones and Wagers, 2008*).

In *C. elegans*, the effects of GSC absence have been studied by laser ablation of GSC precursors, which results in a complete loss of GSCs, or by analysis of genetic mutants in which GSC proliferation is inhibited so that the GSC number is very low, and mature germ cells are not formed (*Hsin and Kenyon, 1999*; *Arantes-Oliveira et al., 2002*; *Kenyon, 2010*). For simplicity, we will refer to each of these types of animals as GSC(−) animals. The lifespan extension seen in GSC(−) animals (GSC(−) longevity) requires the action of several conserved transcription factors in the intestine, the counterpart of the mammalian liver, digestive system, and adipose tissue (*Kenyon, 2010*; *Antebi, 2013*; *Hansen et al., 2013*). DAF-16/FOXO is needed for longevity from GSC ablation or

**eLife digest** Understanding how animals age may help us to prevent age-related or chronic diseases, such as type 2 diabetes and cancer. The tiny nematode worm known as *C. elegans* is widely used as a model to study aging and has enabled researchers to identify factors that can slow down the aging process. Like other animals, these worms contain female and male sex cells that originate from cells called germline stem cells. The normal lifespan of *C. elegans* is less than three weeks, but when the germline stem cells are removed, the worms can live for much longer.

Reproduction requires a lot of energy, which is typically 'stored' in molecules of fat. Animals utilize their fat reserves and release this energy by breaking the fat molecules down into smaller molecules as part of their 'metabolism'. Worms that have had their germline stem cells removed have altered fat metabolism, and it is thought that this may contribute to their increased lifespan. These worms have increased levels of a protein called SKN-1, which alters fat metabolism and helps to protect cells from toxic molecules and other stresses.

SKN-1 works by regulating the activity (or 'expression') of many genes in cells, but it is not clear how this increases the lifespan of the worms. Steinbaugh et al. studied mutant worms that were lacking SKN-1. Unlike normal worms, when the germline stem cells were removed from the mutants, their lifespan did not increase. Further experiments analyzed the genes that are switched on by SKN-1, and identified many that are involved in fat metabolism, in degrading other proteins, and in detoxifying harmful molecules. The experiments also found that SKN-1 reduces the overall amount of fat stored in the body.

Next, Steinbaugh et al. investigated how SKN-1 stops fat from being stored. During reproduction, cells in the gut produce yolk—which is rich in fats—that will be provided to germ cells to nourish the developing embryo. Worms lacking germline stem cells are not able to reproduce, but they continue to make yolk. Steinbaugh et al. found that the build up of the yolk activates SKN-1, which in turn inhibits the further accumulation of fats.

Steinbaugh et al.'s findings show that SKN-1 can be activated by fat molecules and plays a direct role in controlling the amount of fat stored in the body of the worms. A future challenge will be to identify the specific fat molecules that activate SKN-1, which could provide a model for understanding how specific fats in human diets could have wide-ranging health benefits.

reduced insulin/IGF-1 signaling (IIS) but is regulated differently by each pathway (*Lin et al., 2001*; *Libina et al., 2003*; *Kenyon, 2010*). GSC(−) longevity also requires HLH-30/TFEB, PHA-4/FOXA, and the nuclear receptors DAF-12/FXR, NHR-80/HNF4, and NHR-49/PPARα (*Hsin and Kenyon, 1999*; *Goudeau et al., 2011*; *Lapierre et al., 2011*; *O'Rourke and Ruvkun, 2013*; *Ratnappan et al., 2014*). Under most conditions, GSC(−) longevity also depends upon a hormonal signal from the somatic gonad that activates DAF-12 (*Kenyon, 2010*; *Antebi, 2013*). Aside from the identification of mechanisms required for DAF-16 function, we understand little about how GSCs influence these transcription factors (*Kenyon, 2010*; *Antebi, 2013*).

One hallmark of GSC(−) animals is enhancement of both proteostasis and stress resistance. During aging, GSC(−) animals maintain more robust responses to thermal and proteotoxic stress (*Ben-Zvi et al., 2009*). They also exhibit a striking *daf-16*-dependent increase in activity of the proteasome (*Vilchez et al., 2012*), a multisubunit complex which degrades proteins that the ubiquitylation system has tagged for decay (*Glickman and Ciechanover, 2002*; *Goldberg, 2003*). In addition, GSC removal enhances immunity (*Alper et al., 2010*) and boosts oxidative stress resistance through an undetermined DAF-16-independent mechanism (*Libina et al., 2003*).

Another notable characteristic of GSC(−) animals is that many aspects of lipid metabolism are altered. Expression of particular fatty acid (FA) oxidation, FA desaturation, and triglyceride lipase genes is increased, as is total lipase activity (*Wang et al., 2008*; *Goudeau et al., 2011*; *Lapierre et al., 2011*; *McCormick et al., 2012*; *Ratnappan et al., 2014*). Given that lipid catabolism is elevated, it seems paradoxical that GSC(−) animals also exhibit dramatically increased fat accumulation (*O'Rourke et al., 2009*). Interestingly, GSC(−) longevity seems to depend upon particular lipid metabolism processes. Production of the unsaturated free FA (FFA) oleic acid (OA) is required

(*Goudeau et al., 2011*), as are the triglyceride lipases LIPL-4/LIPA and FARD-1/FAR2 (*Wang et al., 2008*; *McCormick et al., 2012*). It is of intense interest to determine whether the fat accumulation seen with GSC ablation might derive from production and storage of particular beneficial fats, or a salutary overall balance of lipid metabolism that is consistent with longevity (*Ackerman and Gems, 2012*; *Hansen et al., 2013*).

The *C. elegans* transcription factor SKN-1 controls a broad detoxification response to oxidative and xenobiotic stress and is orthologous to the mammalian Nrf1/2/3 (NF-E2-related factor) proteins (*An and Blackwell, 2003*; *Oliveira et al., 2009*; *Park et al., 2009*). SKN-1/Nrf proteins have been implicated in longevity from *C. elegans* to rodents (*An and Blackwell, 2003*; *Bishop and Guarente, 2007*; *Leiser and Miller, 2010*; *Sykiotis and Bohmann, 2010*; *Steinbaugh et al., 2012*; *Ewald et al., 2015*). Recent findings raise the question of whether these transcription regulators might also have important functions in lipid homeostasis. SKN-1/Nrf proteins influence expression of lipid metabolism genes (*Oliveira et al., 2009*; *Paek et al., 2012*; *Hayes and Dinkova-Kostova, 2014*; *Tsujita et al., 2014*), and SKN-1 has been linked to fat mobilization under particular starvation or dietary conditions (*Paek et al., 2012*; *Pang et al., 2014*). Mice that lack Nrf1 in the liver develop non-alcoholic fatty liver disease (NAFLD) that progresses to non-alcoholic steatohepatitis (NASH), and Nrf2$^{-/-}$ mice develop NASH on a high-fat diet (*Xu et al., 2005*; *Okada et al., 2013*; *Tsujita et al., 2014*). However, reduced Nrf protein function is thought to predispose to NASH by impairing hepatic stress resistance (*Xu et al., 2005*; *Lee et al., 2013*). An understanding of NAFLD is a high priority, because its incidence is increasing as a sequella of metabolic syndrome (*Cohen et al., 2011*).

Here, we examined the role of SKN-1 in the effects of GSC absence on lifespan, stress resistance, and lipid metabolism. Genetic inhibition of GSCs activates SKN-1, thereby increasing lifespan and stress resistance. Expression profiling revealed that GSC(−) animals upregulate stress defense, extracellular matrix, and lipid metabolism genes, in many cases dependent upon *skn-1*. SKN-1 is also required for GSC inhibition to increase proteasome activity. SKN-1 is needed for GSC(−) longevity but reduces lipid storage, arguing against the idea that GSC(−) animals simply accumulate beneficial fat. Instead, these high-fat levels appear to derive from unconsumed yolk that was produced for reproduction. Unexpectedly, in GSC(−) animals, SKN-1 appears to be activated by specific FA signals, defining a new mechanism of SKN-1/Nrf protein regulation and GSC-to-soma communication. This homeostatic function of SKN-1 in lipid metabolism suggests that Nrf proteins have a similar role in preventing NASH.

## Results

### SKN-1 promotes longevity and stress resistance in the absence of GSCs

To investigate the importance of *skn-1* in GSC(−) animals, we analyzed temperature-sensitive (ts) mutations in *glp-1*/Notch, which is required for GSC proliferation (*Kimble and Crittenden, 2005*). *glp-1(ts)* mutants that undergo larval development at the non-permissive temperature of 25°C (GSC(−) animals) are sterile, exhibit a markedly reduced number of GSCs, and live considerably longer than wild type (WT) controls (*Arantes-Oliveira et al., 2002*) (*Figure 1A,B*). By contrast, this lifespan extension was blocked in a *skn-1* mutant background (*Figure 1A,B*). Lack of *skn-1* also impaired lifespan extension when *glp-1(ts)* animals were downshifted to 20°C after development was complete (*Table 1*). Similar results were obtained with or without 5-fluoro-2′-deoxyuridine (FUdR), which inhibits offspring formation in the control (*Table 1*). Consistent with these findings, in an earlier experiment in which *glp-1(ts)* extended lifespan by less than 7%, *skn-1* knockdown by RNA interference (RNAi) prevented this increase (*Vilchez et al., 2012*). In contrast to *daf-16*, *skn-1* was also required for GSC inhibition to increase oxidative stress resistance (*Figure 1C–F*; *Table 2*).

### GSCs regulate intestinal DAF-16 and SKN-1 through different mechanisms

We investigated whether the benefits of GSC absence simply require that SKN-1 be present or involve activation of SKN-1. SKN-1 accumulates in intestinal nuclei in response to certain stresses, or inhibition of mechanisms that include IIS, mTORC2, glycogen synthase kinase-3, translation elongation, and the ubiquitin ligase WDR-23 (*An and Blackwell, 2003*; *Tullet et al., 2008*; *Choe et al.,*

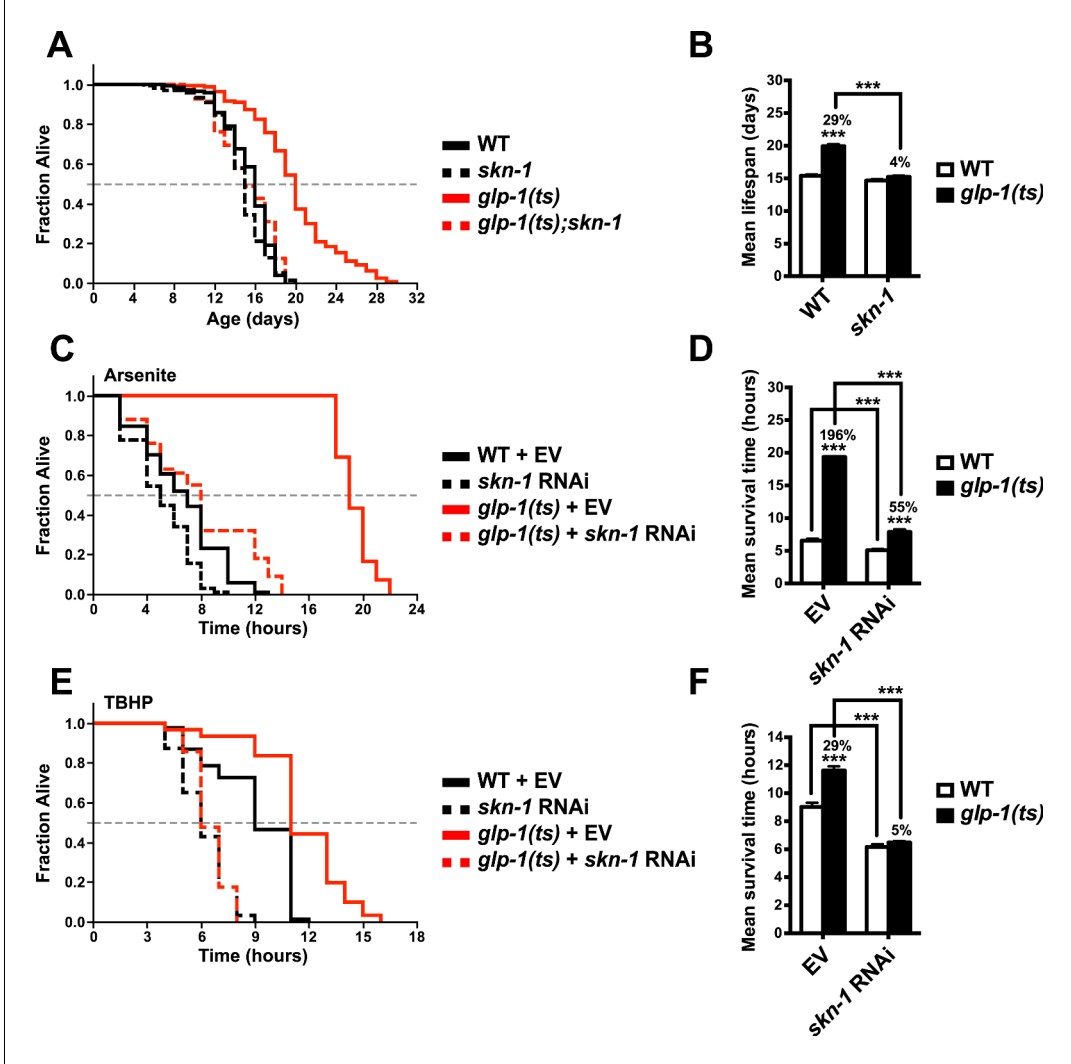

**Figure 1.** SKN-1 promotes longevity and stress resistance in germline stem cell (GSC)(−) animals. (A, B) Wild type, *skn-1(zu135)*, *glp-1(bn18ts)*, and *glp-1(bn18ts);skn-1(zu135)* double mutants were assayed for lifespan at 25°C. *skn-1(zu135)* is a presumed null that is used throughout the study. Unless otherwise specified, *glp-1(ts)* refers to *glp-1(bn18ts)*. (A) Composite survival curve. (B) Graph of mean lifespans. (C–F) *glp-1(ts)* mutants require *skn-1* for oxidative stress resistance. Day-3 adult *glp-1(ts)* and control worms treated with *skn-1* RNAi or empty vector were exposed to (C, D) 5 mM sodium arsenite (AS) or (E, F) 15.4 mM tert-butyl hydroperoxide (TBHP). Data are represented as mean ± SEM. p < 0.001***. The interaction between *glp-1* and *skn-1* was significant for both lifespan and stress resistance (p < 0.001). Statistical analysis and replicates are in *Tables 1*, *2*.

2009; Park et al., 2009; Li et al., 2011; Robida-Stubbs et al., 2012). The levels of a SKN-1::GFP (green fluorescent protein) fusion in intestinal nuclei were also notably elevated in GSC(−) animals (*Figure 2A,B*). This was associated with increased expression of direct SKN-1 target genes, apparently through activation of their intestinal expression (*Figure 2C–E*). The KRI-1/KRIT1 ankyrin repeat protein and the TCER-1/TCERG1 transcription factor are required for GSC absence to induce DAF-16 nuclear accumulation and extend lifespan (*Berman and Kenyon, 2006*; *Ghazi et al., 2009*). In contrast, in GSC(−) animals SKN-1 nuclear accumulation was only partially or minimally affected by loss of *kri-1* or *tcer-1*, respectively, but was abolished by knockdown of the *pmk-1*/p38 kinase (*Figure 2F,G*), which phosphorylates SKN-1 and under most circumstances is required for SKN-1 nuclear accumulation (*Inoue et al., 2005*). In GSC(−) animals, DAF-12 is needed for DAF-16 nuclear accumulation and activity (*Berman and Kenyon, 2006*; *Kenyon, 2010*; *Antebi, 2013*), and induces expression of the microRNAs *mir-84* and *mir-241*, which target inhibitors of DAF-16 (*Shen et al.,*

**Table 1.** Lifespans

| Set | Strain | Mean lifespan ± SEM (days) | Median lifespan (days) | 75th % (days) | N | % Mean lifespan Ext. | p value |
|---|---|---|---|---|---|---|---|
| Lifespan at 25°C | | | | | | | |
| Composite lifespan at 25°C with FUdR (from 2 replicates) | | | | | | | |
| C1 | N2 | 15.40 ± 0.2 | 16 | 17 | 133/148 | – | – |
| | *skn-1(zu135)* | 14.66 ± 0.2 | 15 | 16 | 158/167 | – | – |
| | *glp-1(bn18ts)* | 19.94 ± 0.3 | 20 | 22 | 164/171 | 29.48 | <0.0001* |
| | *glp-1(bn18ts);skn-1(zu135)* | 15.23 ± 0.2 | 15 | 18 | 149/170 | 3.89 | <0.0001† |
| | two-way ANOVA *glp-1(ts)* and *skn-1* interaction | | | | | | <0.0001 |
| Replicate lifespans at 25°C with FUdR | | | | | | | |
| #1 | N2 | 13.99 ± 0.4 | 14 | 17 | 40/55 | – | – |
| | *skn-1(zu135)* | 12.54 ± 0.3 | 13 | 14 | 41/50 | – | – |
| | *glp-1(bn18ts)* | 22.46 ± 0.8 | 25 | 27 | 50/55 | 60.54 | <0.0001* |
| | *glp-1(bn18ts);skn-1(zu135)* | 13.29 ± 0.4 | 12 | 14 | 53/70 | 6.00 | <0.0001† |
| | *glp-1(ts)* and *skn-1* interaction | | | | | | <0.0001 |
| #2 | N2 | 16.07 ± 0.2 | 16 | 17 | 93/93 | – | – |
| | *skn-1(zu135)* | 15.42 ± 0.2 | 15 | 17 | 117/117 | – | – |
| | *glp-1(bn18ts)* | 18.86 ± 0.3 | 19 | 21 | 114/116 | 17.36 | <0.0001* |
| | *glp-1(bn18ts);skn-1(zu135)* | 16.42 ± 0.2 | 17 | 18 | 96/100 | 6.49 | <0.0001† |
| | *glp-1(ts)* and *skn-1* interaction | | | | | | 0.0002 |
| Lifespan at 20°C (25°C during development, downshifted to 20°C at D1 adulthood) | | | | | | | |
| Lifespan at 20°C without FUdR | | | | | | | |
| #3 | N2 | 20.36 ± 0.6 | 18 | 21 | 42/50 | – | – |
| | *skn-1(zu135)* | 18.39 ± 0.5 | 18 | 18 | 27/47 | – | – |
| | *glp-1(bn18ts)* | 25.87 ± 1.3 | 24 | 33 | 35/66 | 27.06 | 0.0002* |
| | *glp-1(bn18ts);skn-1(zu135)* | 20.02 ± 0.6 | 18 | 24 | 34/55 | 8.86 | <0.0001† |
| | *glp-1(ts)* and *skn-1* interaction | | | | | | 0.0230 |
| Composite lifespan at 20°C with FUdR (from 2 replicates) | | | | | | | |
| C2 | N2 | 20.51 ± 0.5 | 21 | 24 | 86/90 | – | – |
| | *skn-1(zu135)* | 17.78 ± 0.4 | 19 | 20 | 88/94 | – | – |
| | *glp-1(bn18ts)* | 24.52 ± 0.6 | 25 | 28 | 81/104 | 19.55 | <0.0001* |
| | *glp-1(bn18ts);skn-1(zu135)* | 20.47 ± 0.5 | 20 | 24 | 91/98 | 15.13 | <0.0001† |
| | *glp-1(ts)* and *skn-1* interaction | | | | | | 0.1892 |
| Replicate lifespans at 20°C with FUdR | | | | | | | |
| #4 | N2 | 18.09 ± 0.4 | 17 | 19 | 33/37 | – | – |
| | *skn-1(zu135)* | 14.78 ± 0.5 | 14 | 17 | 31/35 | – | – |
| | *glp-1(bn18ts)* | 21.79 ± 1.0 | 21 | 28 | 28/50 | 20.45 | 0.0002* |
| | *glp-1(bn18ts);skn-1(zu135)* | 16.83 ± 0.3 | 17 | 18 | 36/43 | 13.87 | <0.0001† |
| | *glp-1(ts)* and *skn-1* interaction | | | | | | 0.1491 |
| #5 | N2 | 22.00 ± 0.6 | 23 | 25 | 53/53 | – | – |
| | *skn-1(zu135)* | 19.40 ± 0.4 | 20 | 21 | 57/59 | – | – |
| | *glp-1(bn18ts)* | 25.91 ± 0.7 | 25 | 30 | 53/54 | 17.77 | <0.0001* |
| | *glp-1(bn18ts);skn-1(zu135)* | 22.80 ± 0.6 | 24 | 26 | 55/55 | 17.53 | <0.0001† |
| | *glp-1(ts)* and *skn-1* interaction | | | | | | 0.6607 |

Percent lifespan extension refers to *glp-1(ts)* vs wild type or *skn-1* control. p values were calculated by log-rank test. Symbols denote effect relative to N2* or *glp-1(ts)*†. The interaction effect of *glp-1(ts)* and *skn-1* was calculated by two-way ANOVA using mean lifespan. The last p value reflects the

specific requirement of *skn-1* for *glp-1(ts)* lifespan, as opposed to its effect on lifespan in general. Homozygous *skn-1* mutants produce eggs that do not hatch because of a catastrophic defect in developmental patterning but do not exhibit known defects in the germline itself (*Bowerman et al., 1992*).

*2012*). In GSC(−) animals, *daf-12* knockdown only mildly affected SKN-1::GFP accumulation (*Figure 2G*), and SKN-1 target gene induction was generally not impaired by *daf-12* or *mir-241;mir-84* mutations (*Figure 2H*). The absence of GSCs therefore activates SKN-1 in the intestine but through a different mechanism from DAF-16.

## SKN-1 reprograms stress resistance and metabolism in GSC(−) animals

To investigate how SKN-1 promotes longevity and stress resistance upon GSC loss, we used RNA sequencing (RNA-seq) to identify genes that are (1) expressed at higher levels in adult somatic tissues when germ cells are largely absent and (2) dependent upon SKN-1 (*Figure 3A*). We compared intact *glp-1(ts)* (GSC(−)) animals to wild-type GSC(+) controls at the non-permissive temperature of 25°, analyzing day-1 adults in which development was complete and performing differential expression analyses on the normalized RNA-seq data for 12,595 expressed genes (*Figure 3B*). We detected similar expression levels of SKN-1 upregulated targets in qRT-PCR analyses of the samples used for sequencing, giving us confidence in our RNA-seq results (*Figure 3—figure supplement 1A*). Moreover, in the GSC(−) gene set, the canonical DAF-16 targets *mtl-1* and *sod-3* (*Murphy et al., 2003*; *Kenyon, 2010*) were upregulated (*Supplementary file 1a*), and functional groups that are characteristic of germline-specific genes were under-represented (*Figure 3—figure supplement 1B*).

No previous studies have globally profiled genes that are upregulated in the soma in response to germ cell loss. mRNAs that are present at higher relative levels in GSC(−) samples compared to WT would include not only those genes, but also genes that are expressed only in somatic cells, because the germline accounts for about two-thirds of all adult nuclei (*Kimble and Crittenden, 2005*). To gauge the maximal extent of this background, we examined mRNAs that are expressed specifically in somatic tissues. These somatic-specific mRNAs were enriched threefold to fourfold in the GSC(−) samples, approximately, the level predicted from the 2:1 proportion of germline to somatic nuclei (*Figure 3—figure supplement 1C*). Accordingly, if an mRNA that is not somatic specific was present at a fourfold-elevated level in GSC(−) samples, we considered this mRNA to be upregulated in the soma in response to GSC absence, although we expect that this stringent cutoff would miss many upregulated genes.

In GSC(−) animals, 1306 and 615 genes were upregulated more than fourfold and fivefold, respectively, indicating a broad remodeling of transcription (*Figure 3—figure supplement 1D,E*; *Supplementary file 1a*). In the latter set, which is more amenable to functional annotation analysis because of its smaller size, the most prominently overrepresented category was collagen (*Figure 3C*; *Supplementary file 1a*). Although collagens may be expressed primarily in the soma, this overrepresentation is likely to be meaningful because of the extent to which these genes were upregulated (*Supplementary file 1a*), and because collagens are generally overrepresented in other longevity-associated gene sets, with certain collagens being critical for lifespan extension (*Ewald et al., 2015*). Also, notably upregulated in GSC(−) animals were genes involved in detoxification, immunity (C-type lectin and galectin), and metabolism, particularly FA oxidation and other lipid metabolism processes (*Figure 3C*; *Supplementary file 1a*).

We used RNAi to investigate the contribution of *skn-1* to gene expression in GSC(−) animals (*Figure 3A*). A previous microarray analysis of *skn-1* RNAi-treated L4 larvae at 20°C found that in the absence of acute stress, SKN-1 upregulates genes involved in processes that include detoxification, lipid metabolism, immunity, and proteostasis (*Oliveira et al., 2009*; *Li et al., 2011*). Similar processes were prominent in the sets of genes for which *skn-1* RNAi reduced expression at 25°C in day-1 WT adults ('SKN-1 upregulated in WT genes'; ≥33% reduction, p < 0.05) (*Supplementary file 1b*) or day-1 GSC(−) animals ('SKN-1-upregulated in GSC(−) genes'; ≥33% reduction, p < 0.05) (*Supplementary file 1c*). Our finding that SKN-1 nuclear occupancy is increased in GSC(−) animals (*Figure 2*) predicts that GSC inhibition would induce SKN-1 to activate genes. Accordingly, *skn-1* RNAi reduced expression of 87 genes that were upregulated at least fourfold in *glp-1(ts)* compared to WT (*Figure 3D,E*; *Supplementary file 1d*). This number probably underestimates the full

**Table 2.** Stress resistance assays

| Set | Strain | Mean survival ± SEM (hrs) | Median survival (hrs) | 75th % survival (hrs) | N | % Mean survival Ext. | p value |
|---|---|---|---|---|---|---|---|
| AS (day 1 adulthood) 25°C continuous; RNAi from L1 | | | | | | | |
| #1 | N2 + vector RNAi | 20.20 ± 1.0 | 22.5 | 22.5 | 122/122 | – | – |
| | N2 + *rme-2* RNAi | 39.51 ± 1.1 | 47.5 | 47.5 | 117/117 | 95.59 | <0.0001* |
| #2 | N2 + vector RNAi | 26.82 ± 1.2 | 28.0 | 28.0 | 57/57 | – | – |
| | N2 + *rme-2* RNAi | 53.21 ± 0.9 | 52.5 | 71.5 | 252/252 | 98.44 | <0.0001* |
| | *glp-1(bn18ts)* + vector RNA | 53.06 ± 1.0 | 52.5 | 71.5 | 216/216 | 97.85 | <0.0001* |
| #3 | N2 + vector RNAi (**20°C**) | 38.59 ± 0.6 | 47.5 | 47.5 | 281/281 | – | – |
| | N2 + *rme-2* RNAi | 63.00 ± 0.7 | 71.5 | 71.5 | 287/287 | 63.25 | <0.0001* |
| #4 | N2 + vector RNAi (**20°C**) | 49.10 ± 0.8 | 46.0 | 64.0 | 306/306 | – | – |
| | N2 + vector/*skn-1* mix RNAi | 32.03 ± 0.6 | 40.0 | 40.0 | 271/271 | – | – |
| | N2 + vector/*rme-2* mix RNAi | 65.38 ± 1.1 | 64.0 | 70.0 | 361/361 | 33.16 | <0.0001* |
| | N2 + *rme-2*/*skn-1* mix RNAi | 33.76 ± 0.5 | 40.0 | 40.0 | 409/409 | 5.40 | <0.0001‡ |
| | two-way ANOVA *rme-2* and *skn-1* interaction | | | | | | <0.0001 |
| #5 | N2 + vector RNAi | 23.18 ± 0.5 | 22.5 | 22.5 | 125/125 | – | – |
| | N2 + *lipl-3* RNAi | 23.04 ± 0.6 | 22.5 | 28.0 | 139/139 | – | – |
| | N2 + *sbp-1* RNAi | 11.01 ± 0.6 | 7.5 | 22.5 | 163/163 | – | – |
| | N2 + *skn-1* RNAi | 20.88 ± 0.7 | 22.5 | 28.0 | 115/115 | – | – |
| | *glp-1(bn18ts)* + vector RNAi | 40.58 ± 1.1 | 47.5 | 47.5 | 105/105 | 75.05 | <0.0001* |
| | *glp-1(bn18ts)* + *lipl-3* RNAi | 28.52 ± 1.0 | 28.0 | 32.5 | 178/178 | 23.77 | <0.0001† |
| | *glp-1(bn18ts)* + *sbp-1* RNAi | 7.50 ± 0.5 | 6.0 | 7.5 | 138/138 | −31.88 | <0.0001† |
| | *glp-1(bn18ts)* + *skn-1* RNAi | 12.90 ± 0.9 | 9.0 | 22.5 | 129/129 | −38.21 | <0.0001† |
| | | | | | | *glp-1(ts)* and *lipl-3* interaction | <0.0001 |
| | | | | | | *glp-1(ts)* and *sbp-1* interaction | <0.0001 |
| | | | | | | *glp-1(ts)* and *skn-1* interaction | <0.0001 |

Table 2 continued

| Set | Strain | Mean survival ± SEM (hrs) | Median survival (hrs) | 75th % survival (hrs) | N | % Mean survival Ext. | p value |
|---|---|---|---|---|---|---|---|
| #6 | N2 + vector RNAi | 24.36 ± 0.8 | 22.5 | 28.0 | 121/121 | – | – |
| | N2 + fat-6/7 mix RNAi | 15.56 ± 0.8 | 22.5 | 22.5 | 159/159 | – | – |
| | N2 + skn-1 RNAi | 20.94 ± 0.6 | 22.5 | 22.5 | 124/124 | – | – |
| | glp-1(bn18ts) + vector RNAi | 38.49 ± 1.5 | 47.5 | 47.5 | 98/98 | 58.03 | <0.0001* |
| | glp-1(bn18ts) + fat-6/7 mix RNAi | 12.04 ± 0.7 | 9.0 | 22.5 | 153/153 | −22.66 | <0.0001† |
| | glp-1(bn18ts) + skn-1 RNAi | 23.69 ± 0.9 | 28.0 | 32.5 | 116/116 | 13.13 | <0.0001† |
| | glp-1(ts) and fat-6/7 interaction | | | | | | <0.0001 |
| | glp-1(ts) and skn-1 interaction | | | | | | <0.0001 |
| AS (day 3 adulthood) 25°C during development, 20°C from D1; RNAi from D1 | | | | | | | |
| #7 | N2 + vector RNAi | 6.55 ± 0.3 | 7.0 | 8.0 | 104/104 | – | – |
| | N2 + skn-1 RNAi | 5.08 ± 0.2 | 5.0 | 7.0 | 103/103 | – | – |
| | glp-1(bn18ts) + vector RNAi | 19.36 ± 0.1 | 19.0 | 20.0 | 97/97 | 195.57 | <0.0001* |
| | glp-1(bn18ts) + skn-1 RNAi | 7.86 ± 0.4 | 8.0 | 12.0 | 100/100 | 54.72 | <0.0001† |
| | glp-1(ts) and skn-1 interaction | | | | | | <0.0001 |
| #8 | N2 + vector RNAi | 8.31 ± 0.4 | 9.0 | 10.0 | 98/98 | – | – |
| | N2 + skn-1 RNAi | 6.89 ± 0.3 | 8.0 | 9.0 | 90/90 | – | – |
| | glp-1(bn18ts) + vector RNAi | 16.95 ± 1.0 | 20.0 | 26.0 | 81/81 | 103.97 | <0.0001* |
| | glp-1(bn18ts) + skn-1 RNAi | 9.90 ± 0.4 | 10.0 | 12.0 | 91/91 | 43.69 | <0.0001† |
| | glp-1(ts) and skn-1 interaction | | | | | | <0.0001 |
| TBHP (day 3 adulthood) 25°C during development, 20°C from D1; RNAi from D1 | | | | | | | |
| #9 | N2 + vector RNAi | 9.02 ± 0.3 | 9.0 | 11.0 | 84/108 | – | – |
| | N2 + skn-1 RNAi | 6.16 ± 0.2 | 6.0 | 7.0 | 63/98 | – | – |
| | glp-1(bn18ts) + vector RNAi | 11.62 ± 0.3 | 11.0 | 13.0 | 61/61 | 28.82 | <0.0001* |
| | glp-1(bn18ts) + skn-1 RNAi | 6.48 ± 0.1 | 8.0 | 7.0 | 63/65 | 5.19 | <0.0001† |
| | glp-1(ts) and skn-1 interaction | | | | | | <0.0001 |
| #10 | N2 | 4.29 ± 0.2 | 4.0 | 4.0 | 65/94 | – | – |
| | skn-1(zu135) | 4.51 ± 0.1 | 5.0 | 3.0 | 73/74 | – | – |
| | glp-1(bn18ts) | 6.25 ± 0.2 | 6.0 | 4.0 | 73/74 | 45.69 | <0.0001* |
| | glp-1(bn18ts);skn-1(zu135) | 4.78 ± 0.1 | 5.0 | 4.0 | 73/75 | 5.99 | <0.0001† |
| | glp-1(ts) and skn-1 interaction | | | | | | <0.0001 |

Survival after sodium arsenite (AS) or tert-butyl hydroperoxide (TBHP) treatment was assayed in adult animals. The increase in oxidative stress resistance of glp-1(ts) germline stem cell (GSC(−)) animals was impaired by loss of fat-6/7, lipl-3, sbp-1, and skn-1. Representative assays are shown. Percent survival extension refers to glp-1(ts) or rme-2 RNAi vs the matching wild type or skn-1 control. p values were calculated by log-rank test. Symbols denote effect relative to N2*, glp-1(ts)†, or rme-2 RNAi‡. The interaction effect of GSC(−) or rme-2 with skn-1, or fat-6/7, lipl-3, and sbp-1 were calculated by two-way ANOVA using mean lifespan. The last p value reflects the specific requirement of each gene for GSC(−) or rme-2 stress resistance as opposed to its effect on stress resistance in general.

contribution of SKN-1, because RNAi only partially reduces its activity. In addition to detoxification and lipid metabolism genes, these 87 genes included many involved in extracellular matrices

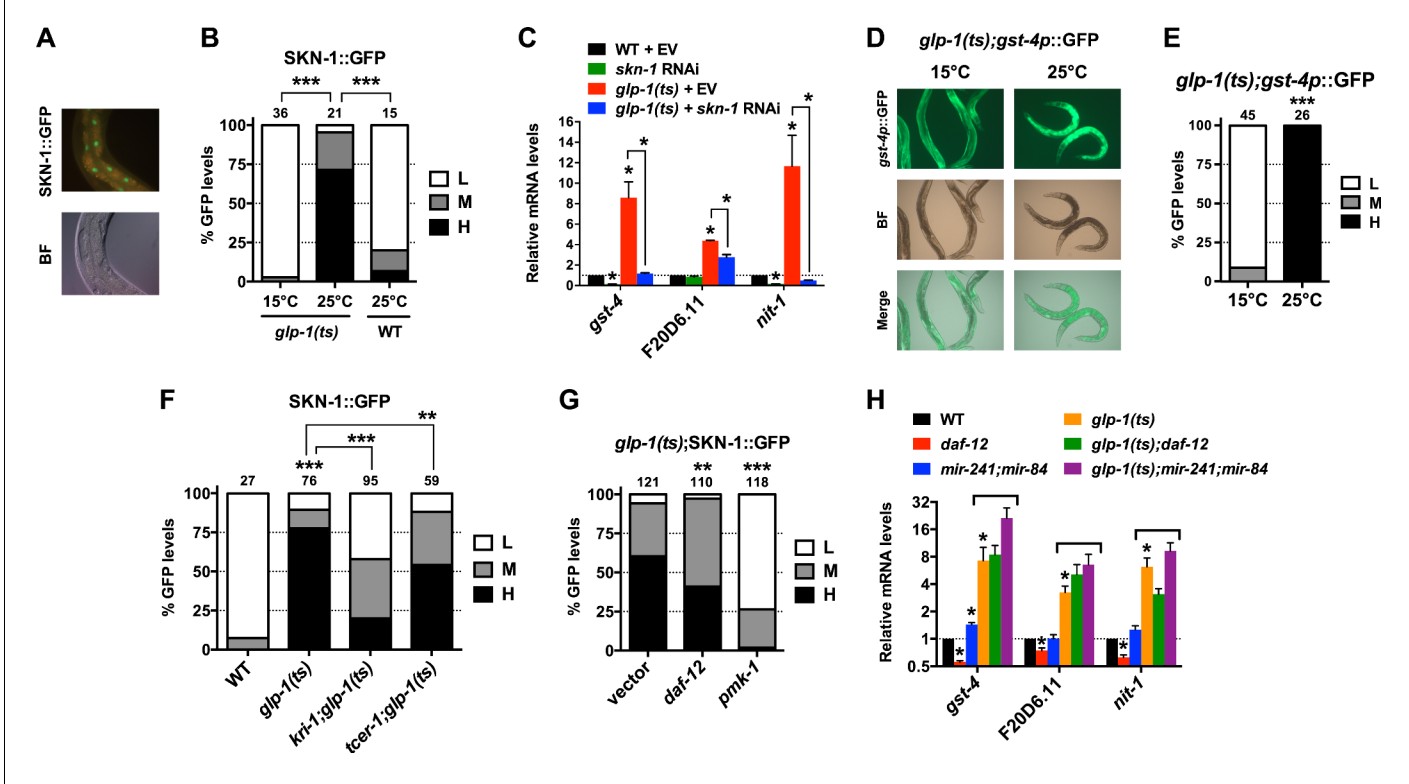

**Figure 2.** GSCs inhibit SKN-1 activity in the intestine. (A) Representative images of SKN-1::green fluorescent protein (GFP) in intestinal nuclei; GFP channel (top), bright field (BF; bottom). (B) Accumulation of SKN-1::GFP in intestinal nuclei in GSC(−) animals. (C) *skn-1*-dependent activation of direct SKN-1 target genes (*Robida-Stubbs et al., 2012*) in response to GSC absence, detected by qRT-PCR. (D, E) Increased expression of *gst-4p*::GFP in the intestine of *glp-1(ts)* animals. Hypodermal *gst-4p*::GFP expression appeared to be unaffected. (D) Representative 10× images. (E) Intestinal *gst-4p*::GFP quantification. (F–H) GSCs regulate SKN-1 parallel to DAF-16 and DAF-12. In (H), SKN-1 target genes are assayed by qRT-PCR. *glp-1(ts)* refers to *glp-1 (e2141ts)*, and horizontal black lines indicate strains lacking GSCs. (C, H) Data are represented as mean ± SEM. n = 3 for qRT-PCR samples. (B, E–G) GFP quantification with high, medium, low scoring. Numbers above bars denote sample size. p < 0.05*; p < 0.01**; p < 0.001***.

(ECMs), as expected from the *skn-1*-dependence of many ECM genes that are upregulated in other long-lived *C. elegans* (*Ewald et al., 2015*). Many of these *skn-1*-dependent GSC(−) genes appear to be direct SKN-1 targets, as predicted by presence of SKN-1 binding sites in their upstream regions and direct binding of SKN-1 in genome-wide chromatin immunoprecipitation (ChIP) surveys (*Supplementary file 1d*). In summary, SKN-1 upregulates numerous genes that are associated with phenotypes seen in GSC-ablated animals, including increased stress resistance, immunity, and longevity, as well as alterations in lipid metabolism.

## SKN-1 increases proteasome activity in response to GSC loss

A previous RNAi experiment suggested that SKN-1 is dispensable for the elevated proteasome activity seen in GSC(−) animals (*Vilchez et al., 2012*). We re-examined this question because SKN-1 maintains proteasome gene expression and intestinal proteasome activity in WT *C. elegans* (*Li et al., 2011*), and because proteasome genes were prominent in the SKN-1-upregulated GSC(−) gene set (*Supplementary file 1c*). The proteasome holocomplex consists of a 20S barrel-like structure in which proteins are degraded, and a 19S regulatory cap that directs ubiquitylated proteins into this structure (*Glickman and Ciechanover, 2002*; *Goldberg, 2003*). In general, and consistent with previous findings (*Vilchez et al., 2012*), the relative levels of proteasome subunit mRNAs were lower in GSC(−) animals (*Figure 4—figure supplement 1A*), possibly because of the lack of germ cells. In both WT and GSC(−) animals, *skn-1* knockdown comparably decreased expression of 19S and 20S proteasome subunit genes (*Figure 4A* and *Figure 4—figure supplement 1A*), the majority of which

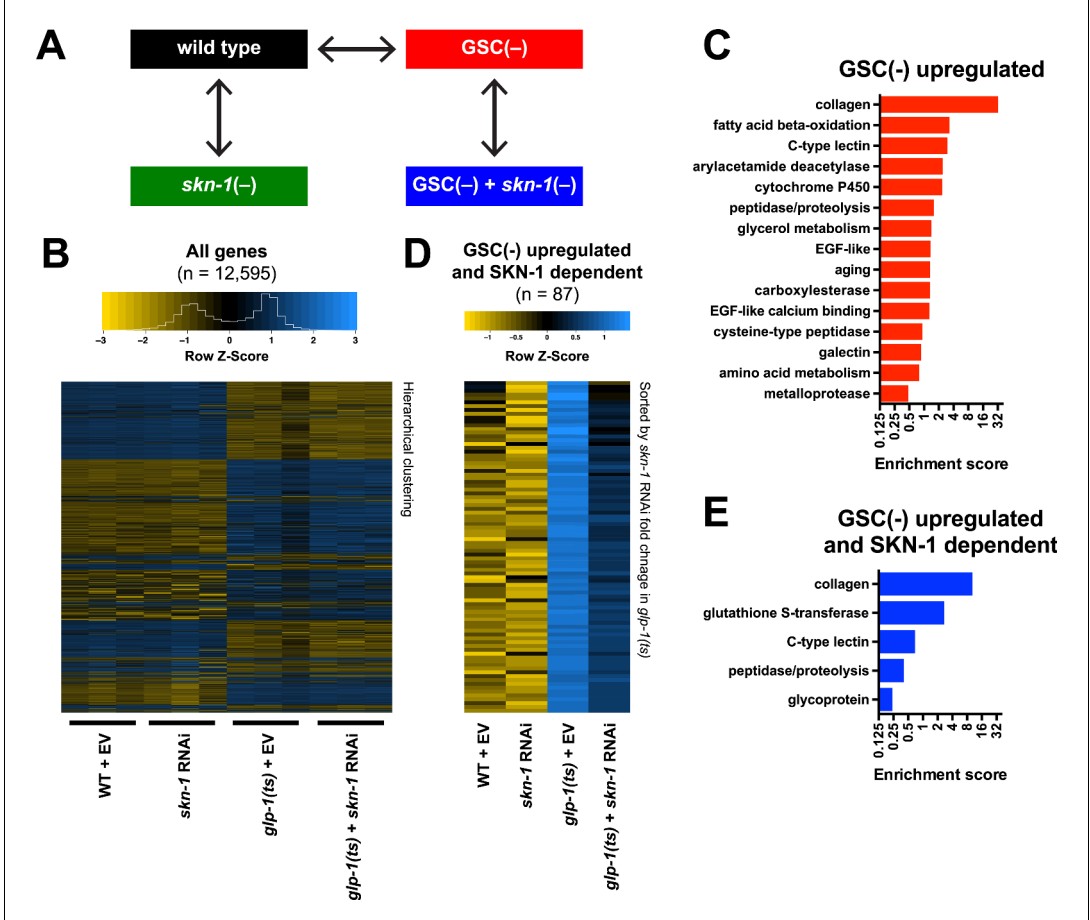

**Figure 3.** Effects of GSC absence and *skn-1* on gene expression. (**A**) RNA-seq experiment setup. For each condition, three biological replicates were obtained from synchronized intact day-1 adults at 25°C. Arrows indicate comparisons that were made, and GSC(−) refers to *glp-1(ts)*. (**B**) Heatmap of all genes evaluated, showing biological replicates. (**C**) DAVID functional annotation analysis of GSC(−)-upregulated genes. (**D**) Heatmap of genes upregulated in *glp-1(ts)* in a *skn-1*-dependent manner (87 genes; fold change (FC) > 4 in GSC(−); FC < 0.67 with *skn-1* RNAi in GSC(−)). (**E**) Functional annotation of the genes shown in (**D**). Genes and additional statistics are provided in *Supplementary file 1a,d*.

The following figure supplement is available for figure 3:

**Figure supplement 1.** Gene expression changes following GSC inhibition.

appear to be direct transcriptional targets of SKN-1 (*Figure 4B*). As these findings would predict, in GSC(−) animals, the lack of *skn-1* dramatically reduced proteasome activity at days 1 and 5 of adulthood (*Figure 4C,D*, and *Figure 4—figure supplement 1B–G*). It is possible that in the earlier analysis (*Vilchez et al., 2012*), RNAi might not have inhibited *skn-1* expression sufficiently to detect its importance for proteasome activity in GSC(−) animals.

The increased proteasome activity of GSC(−) animals is thought to derive from DAF-16-dependent transcriptional upregulation of the RPN-6.1/PSMD11 subunit, which connects the 19S and 20S proteasome particles (*Vilchez et al., 2012*). *rpn-6.1* appears to be unique among proteasome subunit genes, in that its mRNA levels are proportionally higher in GSC(−) animals (*Vilchez et al., 2012*) (*Figure 4E*). *skn-1* was required for this increased *rpn-6.1* expression (*Figure 4E*), and binding site and ChIP studies suggested that *rpn-6.1* is upregulated directly by both SKN-1 and DAF-16 (*Figure 4F*). We conclude that by promoting expression of multiple proteasome subunit genes, including *rpn-6.1*, SKN-1 plays a central role in the increased proteasome activity that results from GSC loss.

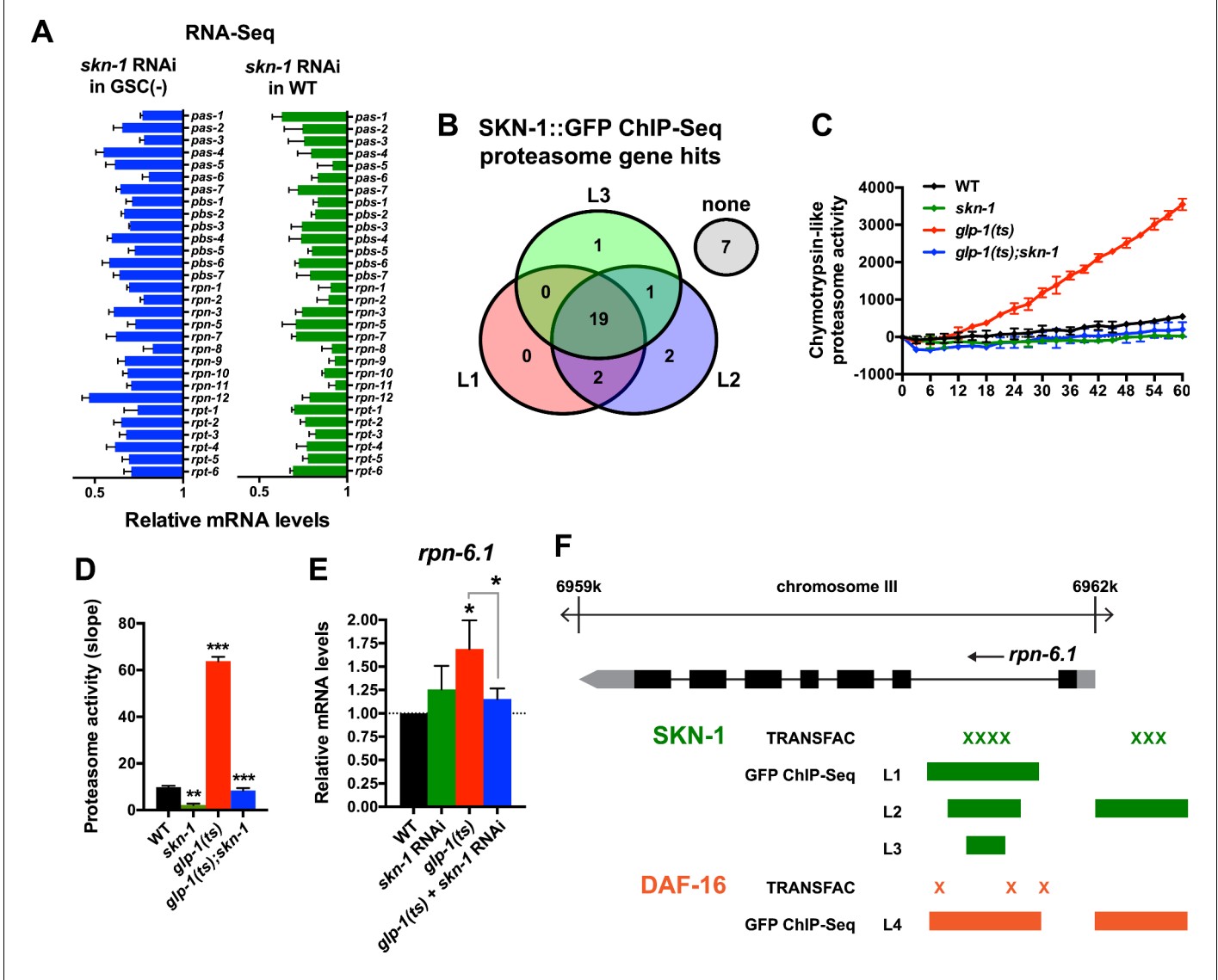

**Figure 4.** SKN-1 increases proteasome activity in response to GSC absence. (A) Reduction in relative proteasome gene subunit mRNA levels by *skn-1* RNAi, detected by RNA-seq. (B) Venn diagram indicating the number of proteasome subunit genes (*pas*, *pbs*, *rpn*, *rpt* families) that have SKN-1::GFP ChIP-seq peak hits near the transcription start site at the indicated larval stage (*Niu et al., 2011*). (C, D) SKN-1-dependence of increased chymotrypsin-related proteasome activity in GSC-ablated worms. The slopes from (C) are graphed in (D). Additional experiments are in *Figure 4—figure supplement 1*. (E) SKN-1-dependence of *rpn-6.1* upregulation. Data are represented as mean ± SEM. n = 3 for all experiments. p < 0.05*; p < 0.01**; p < 0.001***. (F) Direct binding of SKN-1 and DAF-16 to the *rpn-6.1* gene, indicated by TRANSFAC transcription factor binding prediction and modENCODE GFP ChIP-seq analyses. Both the predicted promoter and first intron of *rpn-6.1* are highly enriched for SKN-1 and DAF-16 binding.

The following figure supplement is available for figure 4:

**Figure supplement 1.** SKN-1-dependence of the increased proteasome activity in GSC(−) animals.

## SKN-1 regulates lipid metabolism

Genetic GSC inhibition increased expression of lipid metabolism genes that represent a wide range of processes, including FFA formation from triglyceride lipolysis, as well as FA oxidation, desaturation, and elongation (*Figure 5A*; *Supplementary file 1a*). Many of these genes were also upregulated by SKN-1 (*Figure 5A*; *Supplementary file 1a–c*). Of particular note, the high-confidence GSC

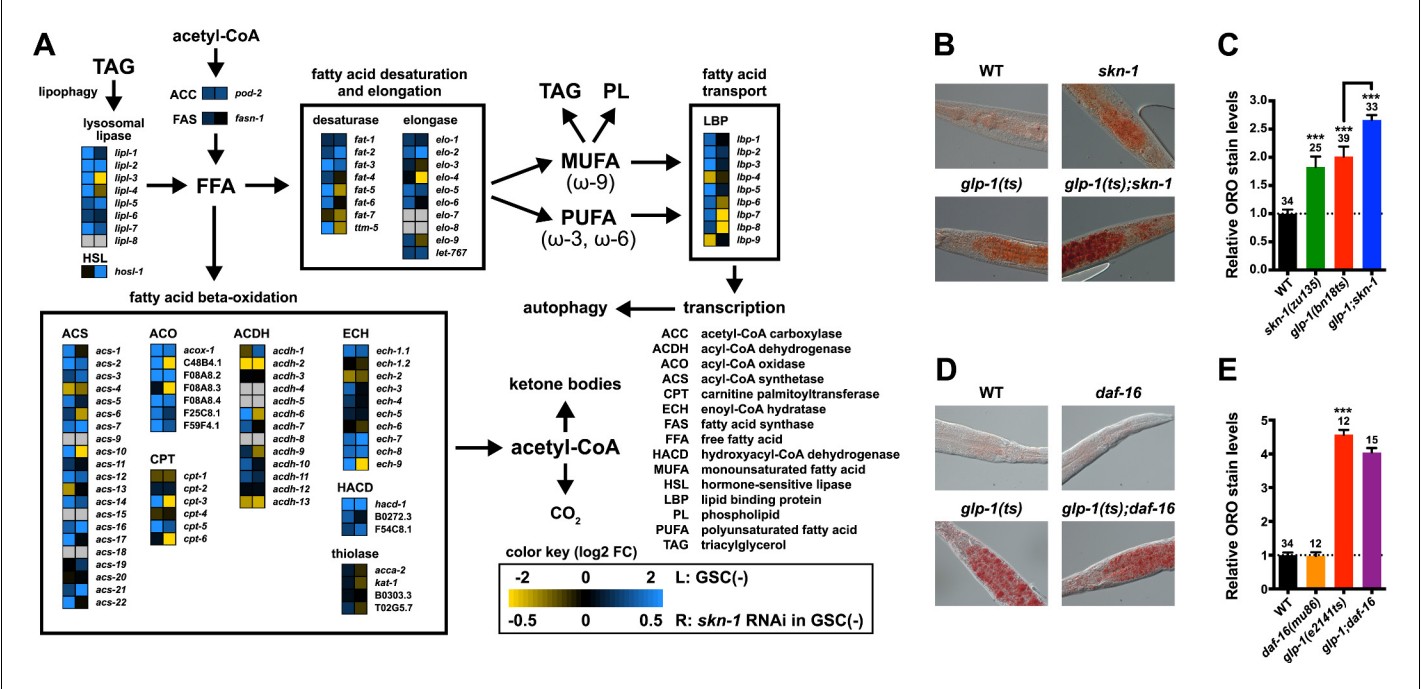

**Figure 5.** SKN-1 regulates lipid metabolism in GSC(−) animals. (**A**) Functional map of lipid metabolism gene expression. Left columns show the effects of GSC absence (GSC(−) vs WT) and right columns the effects of *skn-1* RNAi in GSC(−) animals. SKN-1 regulates genes involved in fatty acid (FA) oxidation, breakdown of triacylglycerols (TAG) to free FAs, production of mono- and poly-unsaturated FAs (MUFA, PUFA), and FA transport. Color coding reflects relative representation in RNA-seq data, with blue and yellow indicating increased and decreased expression, respectively. (**B–E**) Increased fat levels in *glp-1(ts)* and *skn-1* mutants but not *daf-16* mutants. (**D, E**) *glp-1(ts)* refers to *glp-1(e2141ts)*. Representative 40× differential interference contrast (DIC) images of fixed ORO-stained worms are shown in (**B, D**), with quantification provided in (**C, E**). Additional images and quantification are provided in *Figure 5—figure supplement 1*. Data are represented as mean ± SEM. Numbers above bars denote sample size. p < 0.001***.

The following figure supplements are available for figure 5:

**Figure supplement 1.** Representative ORO staining images with quantification.
**Figure supplement 2.** Analysis of the intestinal lipid droplet marker DHS-3::GFP, and TAG levels.
**Figure supplement 3.** RNA-seq counts of select lipid metabolism and yolk transporter genes.

(−) and SKN-1-upregulated gene set included the conserved lysosomal triglyceride lipase *lipl-3*, which increases *C. elegans* lifespan when overexpressed and is normally induced by fasting (*O'Rourke and Ruvkun, 2013*). This gene set also included the FA oxidation genes *acs*-10 (acyl-CoA synthetase), *cpt-3* (carnitine palmitoyltransferase), and *ech-9* (enoyl-CoA hydratase) (*Figure 5A*; *Supplementary file 1d*). This suggested that SKN-1 might have a major role in lipid metabolism, and how it is influenced by GSC absence.

Given that SKN-1 increases both lifespan and stress resistance in GSC(−) animals, its effects on lipid metabolism should also be beneficial. If the elevated fat storage in GSC(−) animals reflects simply elevated production and storage of 'good' lipids, we might expect *skn-1* to support this fat production. We investigated whether SKN-1 affects fat storage in WT and GSC(−) animals by oil red O (ORO) staining of fixed animals, a method that reliably indicates fat accumulation (*O'Rourke et al., 2009*). Remarkably, ablation of *skn-1* by either mutation or RNAi significantly increased lipid levels in either WT or GSC(−) day-1 adults, so that *glp-1(ts);skn-1(−)* animals exhibited markedly high levels of ORO staining (*Figure 5B,C*, and *Figure 5—figure supplement 1A,B*). As an independent method of assessing fat accumulation in the intestine, we examined levels of the predicted short-chain FA

dehydrogenase DHS-3 (*Zhang et al., 2012*). Proteomic and microscopy analyses have shown that DHS-3 localizes almost exclusively to intestinal lipid droplets (*Figure 5—figure supplement 2A*) and marks the vast majority of these lipid droplets in vivo (*Zhang et al., 2012*; *Na et al., 2015*). Consistent with ORO staining, lack of *skn-1* increased accumulation of a DHS-3::GFP fusion in the intestine in WT and GSC(−) animals, without affecting expression of the *dhs-3* mRNA (*Figure 5—figure supplements 2B,C*, *3*). An analysis of total triglyceride levels also indicated that SKN-1 reduces the overall level of fat accumulation (*Figure 5—figure supplement 2D*).

DAF-16 increases lipid accumulation in response to reduced IIS, and influences expression of some lipid metabolism genes in response to GSC removal (*Wang et al., 2008*; *McCormick et al., 2012*). However, consistent with a previous study (*O'Rourke et al., 2009*), we found that loss of *daf-16* did not substantially affect overall fat storage in GSC(−) animals (*Figure 5D,E*, and *Figure 5—figure supplement 1C*). Together, our data indicate that SKN-1 is required to prevent excess fat accumulation under normal feeding conditions, and that SKN-1 but not DAF-16 reduces the lipid load that accumulates in response to GSC loss.

## GSC loss activates SKN-1 through lipid signaling

Given that SKN-1 inhibits fat storage, we considered whether the SKN-1 activation seen in GSC(−) animals might be triggered by lipid accumulation. It is possible that GSC loss simply increases production of certain fats. However, GSC ablation or inhibition prevents formation of oocytes, which endocytose lipid-rich yolk that is synthesized in the intestine (*Grant and Hirsh, 1999*). Fat storage might therefore be increased indirectly by GSC loss, through accumulation of unused yolk lipids. We tested a key prediction of this model by examining yolk accumulation and distribution, which can be visualized with GFP-tagged vitellogenin (YP170/VIT-2::GFP), a major yolk lipoprotein (*Grant and Hirsh, 1999*). VIT-2::GFP was visible primarily in oocytes and embryos in WT day-1 adults but accumulated to extremely high levels throughout the body cavity in the absence of GSCs (*Figure 6A,B*, and *Figure 6—figure supplement 1*). Apparently, yolk production was not slowed sufficiently to compensate for the lack of gametogenesis. The failure to consume yolk-associated lipid could account for the increase in overall fat storage seen in GSC(−) animals.

We investigated whether accumulation of yolk-associated lipids might induce SKN-1 to mount a protective response. Supporting this idea, when the oocyte-specific yolk receptor *rme-2* is knocked down, yolk accumulates to high levels (*Grant and Hirsh, 1999*), and in the intestine, SKN-1 accumulates in nuclei and its target gene *gst-4* is activated (*Figure 6C–F*). Additionally, *rme-2* RNAi increased stress resistance in a *skn-1*-dependent manner (*Figure 6G* and *Table 2*). When de novo lipogenesis was prevented by knockdown of the SREBP1 ortholog *sbp-1* (*Yang et al., 2006*), SKN-1::GFP failed to accumulate in intestinal nuclei in response to GSC inhibition (*Figure 6H,I*), but not oxidative stress (*Figure 6—figure supplement 2A*) or reduced IIS (*daf-2* mutants, *Figure 6—figure supplement 2B*). The *sbp-1* lipogenesis defect can be rescued by supplementation with 600 μM OA (*Yang et al., 2006*), which is the most abundant FA in olive oil, chicken egg yolk, and human adipose tissue (*National Research Council, 1976*; *Kokatnur et al., 1979*). In *C. elegans*, the abundance of OA is increased in GSC(−) animals, and its synthesis by the FA desaturases, FAT-6 and FAT-7 (SCD orthologs), is required for GSC(−) lifespan extension (*Goudeau et al., 2011*). *fat-6/7* were also required for SKN-1 to accumulate in nuclei after GSC inhibition (*Figure 6J*). Moreover, in GSC(−) animals subjected to *sbp-1* RNAi, SKN-1 nuclear accumulation was fully restored by OA supplementation (*Figure 6H,I*). Consistent with their importance for SKN-1 function, *sbp-1* and *fat-6/7* were required for GSC absence to increase stress resistance (*Figure 6—figure supplement 3*).

Together, our data suggest that certain unsaturated lipids are required for SKN-1 to be activated in response to GSC loss, but not necessarily by other stimuli, and therefore, that lipid accumulation per se might activate SKN-1. Accordingly, feeding of either OA- or coconut oil (CO)-activated *gst-4* in WT animals in a *skn-1*-dependent manner without impairing development or reproduction (*Figure 6—figure supplement 2C–E*). Under these conditions, CO feeding provided OA (300 μM with 0.1% CO) along with multiple saturated FAs. CO feeding strongly induced nuclear accumulation of SKN-1 but not DAF-16 (*Figure 6—figure supplement 2F,G*), and *sbp-1* RNAi did not impair DAF-16 nuclear accumulation in GSC(−) animals (*Figure 6—figure supplement 2H,I*), supporting the notion that GSCs regulate SKN-1 and DAF-16 differently.

The lipid overload that results from reproductive failure might induce stress that activates SKN-1. Oxidative stress induced by sodium arsenite (AS) robustly activates the p38/PMK-1 kinase through

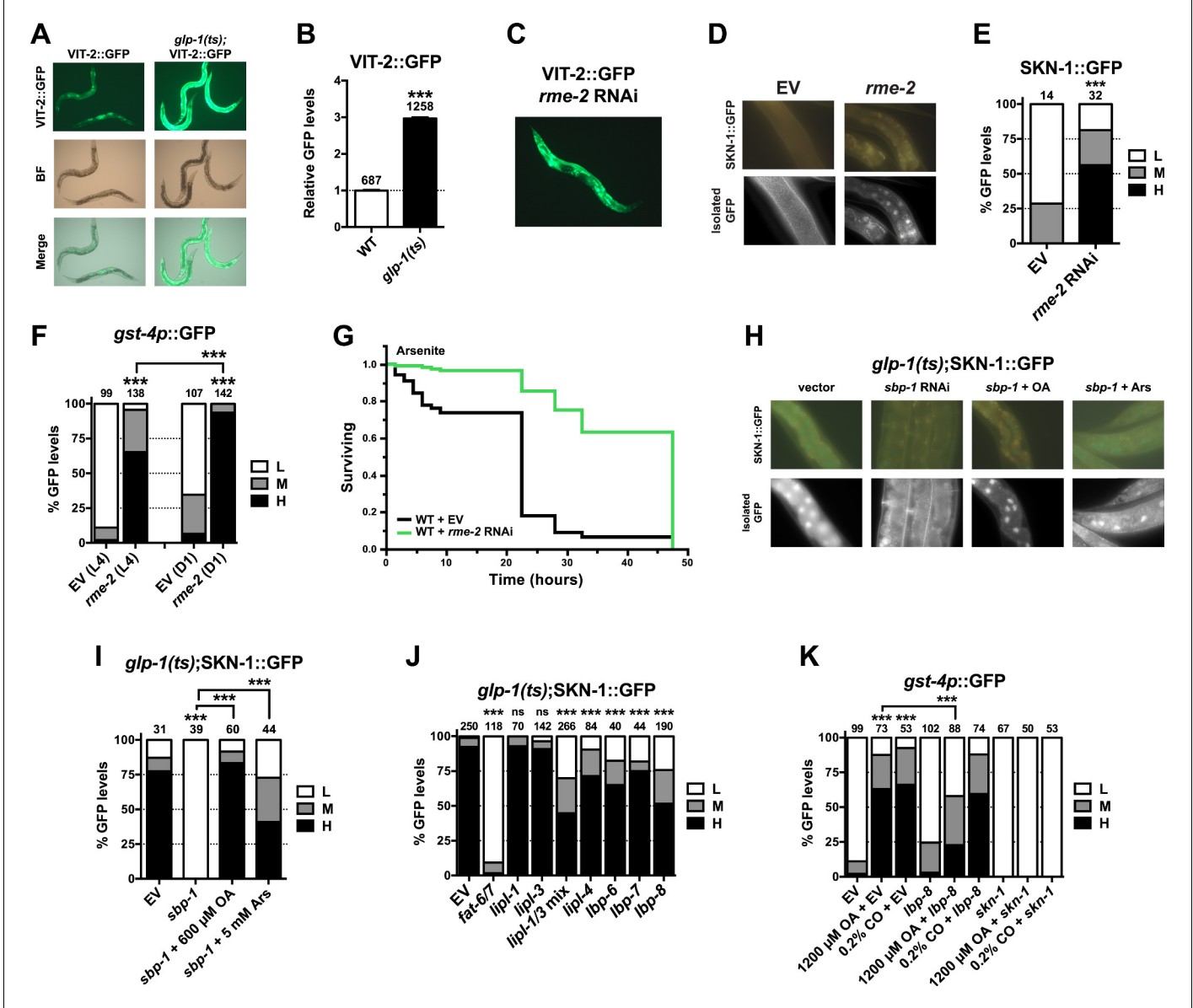

**Figure 6.** GSC absence activates SKN-1 through FA signaling. (**A**) Accumulation of yolk transporter VIT-2::GFP in the soma of GSC(−) animals. Detailed higher magnification images are provided in *Figure 6—figure supplement 1*. (**B**) COPAS quantification of VIT-2::GFP. Data are represented as mean ± SEM. (**C**) Knockdown of the oocyte-specific yolk receptor *rme-2* promotes somatic VIT-2 accumulation. (**D–F**) *rme-2* RNAi activates SKN-1 in the intestine. (**D**, **E**) SKN-1::GFP accumulates in intestinal nuclei in *rme-2* RNAi-treated worms. (**F**) In *rme-2* RNAi-treated worms, *gst-4p*::GFP levels in the intestine are increased at the L4 stage and increased further by day-1 adulthood. (**G**) *rme-2* RNAi enhances resistance to AS, in a *skn-1*-dependent manner (see replicates in *Table 2*). (**H**, **I**) An oleic acid (OA)-dependent signal is required for SKN-1 to be activated by GSC inhibition but not oxidative stress. In GSC(−) animals, SKN-1 nuclear accumulation is abolished by *sbp-1* RNAi and rescued by OA supplementation. SKN-1 remains capable of responding to oxidative stress (30 min AS exposure) after *sbp-1* RNAi in GSC(−) (**H**, **I**) or WT (*Figure 6—figure supplement 2A*) worms. (**J**) Dependence of SKN-1::GFP accumulation in GSC(−) animals on FAT-6/7-mediated FA desaturation, and proteins that generate free unsaturated FAs (LIPL-1/-3 lipases), or transport them to the nucleus (LBP-6/7/8). (**K**) OA and coconut oil (CO) increase *skn-1*-dependent *gst-4p*::GFP expression in the intestine. *lbp-8* RNAi reduces induction by OA. (**A**, **C**) Representative 10× GFP images. (**D**, **H**) Representative 40× GFP images of day-1 adults. (**E**, **F**, **I**–**K**) GFP quantification with high, medium, low scoring. Numbers above bars denote sample size. p < 0.001***.

The following figure supplements are available for figure 6:

**Figure supplement 1.** Enlarged VIT-2::GFP images.

**Figure supplement 2.** SKN-1 is activated in response to FA signaling.

*Figure 6 continued on next page*

Figure 6 continued

**Figure supplement 3.** FA desaturation is required for GSC(−) stress resistance.

phosphorylation, leading in turn to SKN-1 activation (*Inoue et al., 2005*). GSC loss induced SKN-1 nuclear accumulation at least as dramatically as AS treatment (*Figure 2B* and *Figure 6—figure supplement 2A*) but did not detectably increase PMK-1/p38 activity (*Figure 6—figure supplement 2J*), suggesting that any stress arising from the lack of GSCs might not be sufficient on its own to explain SKN-1 activation.

By breaking down triglycerides, the lysosomal lipases LIPL-1/3 and LIPL-4 enable production of specific unsaturated FFAs that promote autophagy and longevity (*Lapierre et al., 2011*; *O'Rourke et al., 2013*; *O'Rourke and Ruvkun, 2013*; *Folick et al., 2015*). Some of these FAs are escorted from the lysosome to the nucleus by the conserved lipid-binding protein LBP-8/FABP1 (*Folick et al., 2015*; *Han and Brunet, 2015*). In GSC(−) animals, SKN-1 nuclear accumulation was inhibited modestly by *lipl-4* RNAi but more strongly by *lipl-1/3* double knockdown (*Figure 6J*). Furthermore, *lipl-3* RNAi reduced stress resistance in GSC(−) but not WT animals (*Figure 6—figure supplement 3A,B*). Given that fat storage is increased in *lipl-1/3* mutants (*O'Rourke and Ruvkun, 2013*), our data suggest that in GSC(−) animals, SKN-1 activity may depend upon particular *lipl-1/3*-dependent products, not lipid levels per se. Knockdown of *lbp-8* or other LBPs also interfered with SKN-1::GFP nuclear accumulation in GSC(−) animals, and *lbp-8* RNAi impaired SKN-1-dependent *gst-4* activation by OA, indicating involvement of FA transport (*Figure 6J,K*). Together, our data suggest that in GSC(−) animals, excessive lipid levels lead to production of OA- and LIPL-1/3-dependent FAs that activate SKN-1, possibly through FA-based signaling (*Figure 7*).

## Discussion

The question of how events in one tissue can influence aging in others is of fundamental importance. The effects of GSC loss in *C. elegans* provide a paradigm for investigating this problem, as well as interactions between a stem cell population and its environment. Here, we determined that GSC inhibition leads to a broad transcriptional reprogramming in somatic tissues that involves activation of SKN-1, and that SKN-1 is required for many beneficial effects of GSC absence, including lifespan extension. Previous work showed that SKN-1 is required for lifespan to be extended by reduced insulin/IGF-1, mTORC1, or mTORC2 signaling, by dietary restriction and by low-level mitochondrial reactive oxygen species (ROS) production (*Bishop and Guarente, 2007*; *Tullet et al., 2008*; *Robida-Stubbs et al., 2012*; *Zarse et al., 2012*; *Schmeisser et al., 2013*; *Mizunuma et al., 2014*; *Moroz et al., 2014*; *Ewald et al., 2015*). Our new data further support the idea that SKN-1/Nrf proteins are broadly important for longevity assurance.

One major role of SKN-1 in GSC(−) animals is to increase proteasome activity (*Figure 4C,D*, and *Figure 4—figure supplement 1*). In WT animals, SKN-1 activates most proteasome subunit genes when the proteasome is inhibited (*Li et al., 2011*). This compensatory function is conserved in its mammalian ortholog Nrf1, which is cleaved and activated when proteasome activity is low (*Radhakrishnan et al., 2010*; *Steffen et al., 2010*; *Radhakrishnan et al., 2014*; *Sha and Goldberg, 2014*). By contrast, in GSC(−) animals, proteasome activity is elevated (*Figure 4C,D*) (*Vilchez et al., 2012*), suggesting that additional mechanisms influence SKN-1/Nrf regulation of proteasome genes. In GSC(−) animals, most if not all proteasome subunit genes are dependent upon SKN-1 for their expression (*Figure 4A*), and SKN-1 and DAF-16 together activate the proteasome subunit gene *rpn-6.1* (*Figure 4E,F*), the levels of which are rate limiting for proteasome activity (*Vilchez et al., 2012*). Overexpression of either *rpn-6.1* or the 20S proteasome subunit *pbs-5* increases *C. elegans* lifespan, and in the latter case, lifespan extension was shown to be *skn-1*-dependent (*Vilchez et al., 2012*; *Chondrogianni et al., 2015*), suggesting that enhancement of proteasome activity may be an important mechanism through which SKN-1/Nrf promotes longevity.

The evidence that GSC(−) longevity is associated with fat accumulation, and altered lipid metabolism has raised the intriguing possibility that GSC absence induces production of lipids that promote

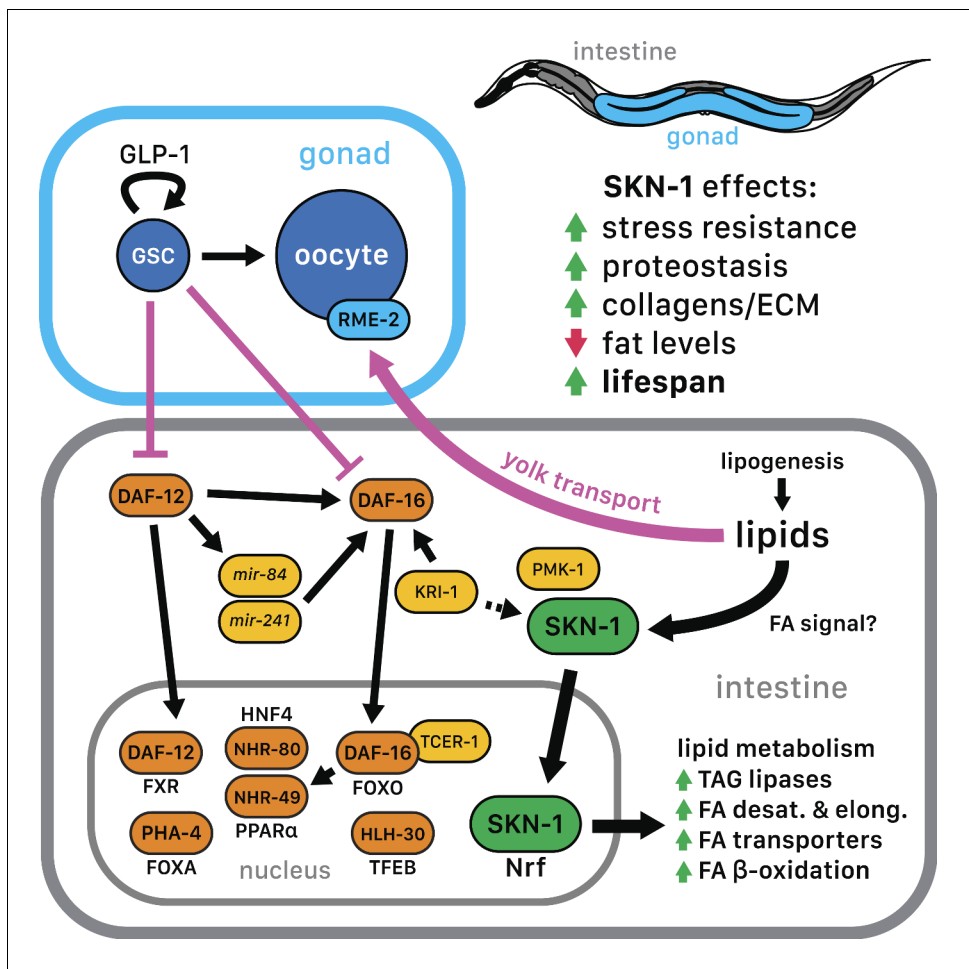

**Figure 7.** SKN-1 regulation in the GSC longevity pathway. GSC absence results in activation of transcription factors in the intestine, with SKN-1 being regulated in parallel to DAF-12 and DAF-16. Yolk transport to oocytes is disrupted by GSC loss, resulting in lipid accumulation in the intestine and body cavity. The resulting SKN-1 activation requires OA, the FAT-6/7 FA desaturases, and the lysosomal lipases LIPL-1/3. This lipid-based signaling to SKN-1 depends partially upon LBP-8, which transports FAs from the lysosome to the nucleus. SKN-1 induces transcription of genes involved in stress resistance, detoxification, proteasome maintenance, extracellular matrix, and lipid metabolism, thereby reducing fat storage and increasing stress resistance and lifespan. Magenta denotes processes that are active in the presence of GSCs.

health and longevity (see 'Introduction'). Our results are consistent with aspects of this model but suggest an important modification. GSC(−) animals accumulate dramatically high levels of yolk lipoproteins by the first day of adulthood (*Figure 6A,B*), providing a likely reason that they accumulate so much lipid. Moreover, SKN-1 acts to *reduce* fat accumulation but is critical for the benefits of GSC loss (*Figure 5B,C*, and *Figure 5—figure supplement 1A,B*), suggesting that GSC(−) animals do not simply overproduce healthful lipids. Finally, excess yolk accumulation induced by another method (*rme-2* RNAi) leads to increased SKN-1 nuclear accumulation and target gene activation, and *skn-1*-dependent stress resistance (*Figure 6D–G*). Taken together, our data suggest that GSC(−) animals accumulate excess fat because they cannot stop production of fat that would otherwise support reproduction (*Figure 7*). Importantly, however, in responding to and metabolizing this fat they produce specific lipids that activate SKN-1 and other regulators, which in turn increase lifespan and may promote a more healthy balance of lipids (*Figure 7*).

Several lines of evidence support this model. GSC inhibition induces SKN-1 and NHR-49 to upregulate largely distinct sets of FA oxidation genes (*Figure 5A*) (*Ratnappan et al., 2014*). This effect of SKN-1 could account for its inhibitory role in fat accumulation (*Figure 5B,C*). A need to

metabolize excess fat could also explain the importance of lipophagy in GSC(−) longevity (*Lapierre et al., 2011*; *Hansen et al., 2013*). With respect to signaling lipids, GSC(−) longevity requires the triglyceride lipase LIPL-4 (*Wang et al., 2008*), which generates unsaturated FFAs that promote longevity (*O'Rourke et al., 2013*; *Folick et al., 2015*). While LIPL-4-dependent FAs act through NHR-49 and NHR-80 (*Lapierre et al., 2011*; *Folick et al., 2015*), and possibly not SKN-1 (*Figure 6J*), in GSC(−) animals, SKN-1 activation involves the LIPL-1/3 lipases (*Figure 6J*), which also promote longevity (*O'Rourke and Ruvkun, 2013*). This elevated SKN-1 activity also depends upon lipid transfer proteins, as well as OA (*Figure 6H–J* and *Figure 6—figure supplement 1C*). OA is required for GSC(−) longevity (*Goudeau et al., 2011*) and is a precursor to unsaturated FAs that have signaling functions (*O'Rourke et al., 2013*; *Folick et al., 2015*). Finally, SKN-1 upregulates *lipl-3* and *lbp-8* in WT and GSC(−) animals (*Figure 5A* and *Figure 5—figure supplement 3*; *Supplementary file 1a,c*), suggesting that it may function both downstream and upstream of lipid signals. The idea that SKN-1 can be activated by lipids that arise from prevention of reproduction and yolk consumption should be considered in evaluation of genetic or pharmacological interventions that increase SKN-1 activity.

SKN-1 is activated by lipids and regulates lipid metabolism gene expression not only in the high-fat GSC(−) model but also in WT animals under normal feeding conditions (*Figures 5*, *6*; *Supplementary file 1b–d*). Moreover, SKN-1 profoundly reduced fat storage by the beginning of adult life in healthy, reproductively active animals that have not begun to age. Taken together, our findings suggest that SKN-1 plays an integral and direct role in maintaining lipid homeostasis. Our results predict that insufficient mammalian Nrf function does not lead to NAFLD/NASH simply by increasing chronic hepatic stress (*Xu et al., 2005*; *Lee et al., 2013*), and that a protective function of Nrf proteins in fat metabolism is likely to be involved. Nrf proteins therefore may provide an important line of defense against metabolic disease. Development of NAFLD is a growing obesity-related public health issue (*Cohen et al., 2011*). Our data suggest that analysis of Nrf proteins could be a promising underexplored direction for investigating causes and prevention of NAFLD, and that SKN-1 and *C. elegans* provide a genetically tractable model that will be valuable in this effort.

Our evidence that GSCs activate SKN-1 through lipid-based signaling suggests a new mechanism through which GSCs influence the soma. The response to GSC loss therefore involves metabolic signals that reflect the altered nutritional balance within the organism. Similar interactions could be important in other stem cell contexts. For example, in the mammalian bone marrow microenvironment, adipose tissue profoundly influences the function of hematopoietic and mesenchymal stem cells (*Adler et al., 2014*; *Muruganandan and Sinal, 2014*). In addition to the mechanisms we have described here, GSC(−) longevity involves endocrine signals from the somatic gonad and depends upon absence of GSCs per se, not simply reproductive cessation (*Kenyon, 2010*; *Antebi, 2013*). It will now be of interest to determine how these mechanisms, as well as other transcription factors that are required for GSC(−) longevity (see 'Introduction'), interface with SKN-1 and its regulation.

Signaling lipids from endogenous, dietary, or microbiota sources constitute an area of considerable excitement, because these signals can induce beneficial effects such as anti-inflammatory protection, enhanced insulin sensitivity, protection against metabolic disease, and increased lifespan in *C. elegans* (*Wang et al., 2008*; *Kniazeva and Han, 2013*; *Lim et al., 2013*; *O'Rourke and Ruvkun, 2013*; *Folick et al., 2015*; *Han and Brunet, 2015*). Signals derived from OA are of particular interest in this regard, because of its dietary availability in olive oil. Mechanisms through which lipid signals are known to act on transcription networks include binding to nuclear receptors, and OA-induced protein kinase A activation that ultimately leads to FA oxidation (*Fu et al., 2003*; *Lim et al., 2013*; *Folick et al., 2015*). By revealing SKN-1 as a new regulator of metabolism and stress defenses that is activated in response to lipids, our data suggest the exciting possibility that this might also be the case for mammalian Nrf proteins. They also suggest that a 'lipohormesis' pathway in which signaling lipids confer health benefits by activating SKN-1/Nrf may not only be a characteristic of GSC-ablated animals but also might be more broadly applicable for enhancing health- and possibly lifespan.

**Table 3.** *C. elegans* strains used in this study

| Code | Genetic background | Transgene | Reference |
|---|---|---|---|
| AA003 | *daf-12(rh61rh411)* X | – | (*Shen et al., 2012*) |
| AA983 | *glp-1(e2141ts)* III; *daf-12(rh61rh411)* X | – | (*Shen et al., 2012*) |
| AA1049 | *mir-241(n4315)* V; *mir-84(n4037)* X | – | (*Shen et al., 2012*) |
| AA1709 | *glp-1(e2141ts)* III; *mir-241(n4315)* V; *mir-84(n4037)* X | – | (*Shen et al., 2012*) |
| AA2735 | *glp-1(e2141ts)* III | – | (*Shen et al., 2012*) |
| CF1903 | *glp-1(e2141ts)* III | – | (*Berman and Kenyon, 2006*) |
| CF1935 | *daf-16(mu86)* I; *glp-1(e2141ts)* III | muIs109[*daf-16p::GFP:: DAF-16 + odr-1p::RFP*] X | (*Berman and Kenyon, 2006*) |
| CL2166 | N2 | dvIs19[pAF15 (*gst-4p::GFP::NLS*)] III | (*Link and Johnson, 2002*) |
| DH1033 | *sqt-1(sc103)* II | bIs1[*vit-2p::VIT-2:: GFP + rol-6(su1006)*] X | (*Grant and Hirsh, 1999*) |
| EU31 | *skn-1(zu135)* IV | – | (*Bowerman et al., 1992*) |
| LD001 | N2 | ldIs7[*SKN-1b/c:: GFP + rol-6(su1006)*] | (*An and Blackwell, 2003*) |
| LD002 | N2 | ldIs1[*SKN-1b/c:: GFP + rol-6(su1006)*] | (*An and Blackwell, 2003*) |
| LD1025 | *daf-2(e1370)* III | ldIs7[*SKN-1b/c:: GFP + rol-6(su1006)*] | (*Tullet et al., 2008*) |
| LD1425 | *glp-1(bn18ts)* III | ldIs1[*SKN-1b/c:: GFP + rol-6(su1006)*] | This study |
| LD1434 | *glp-1(bn18ts)* III; *skn-1(zu135)* IV | – | This study |
| LD1473 | *kri-1(ok1251)* I; *glp-1(bn18ts)* III | ldIs1[*SKN-1b/c:: GFP + rol-6(su1006)*] | This study |
| LD1474 | *tcer-1(tm1452)* II; *glp-1(bn18ts)* III | ldIs1[*SKN-1b/c:: GFP + rol-6(su1006)*] | This study |
| LD1548 | N2 | Is[*dhs-3p::DHS-3:: GFP*] I | (*Zhang et al., 2012*) |
| LD1549 | *glp-1(bn18ts)* III | Is[*dhs-3p::DHS-3:: GFP*] I | This study |
| LD1644 | *sqt-1(sc103)* II; *glp-1(bn18ts)* III | bIs1[*vit-2p::VIT-2:: GFP + rol-6(su1006)*] X | This study |
| LD1653 | *glp-1(bn18ts)* III | – | (*Dorsett et al., 2009*) Outcrossed from DG2389 |
| LD1744 | *glp-1(bn18ts)* III | ldEx119[pAF15 (*gst-4p::GFP::NLS*) + *rol-6(su1006)*] | This study |
| TJ356 | N2 | zIs356[*daf-16p:: DAF-16a/b::GFP + rol-6*] IV | (*Henderson and Johnson, 2001*) |

# Materials and methods

## Strains

Worms were maintained on nematode growth medium (NGM) plates seeded with *Escherichia coli* (OP50) at 15°C, using standard techniques (*Brenner, 1974*). In all experiments, *glp-1(ts)* mutants

were matched with the wild-type N2 strain used for outcrossing. The *C. elegans* strains used in this study are detailed in *Table 3*.

## RNAi

Feeding RNAi was performed using tetracycline-resistant HT115 bacteria carrying the pL4440 plasmid with ampicillin/carbenicillin resistance (*Kamath et al., 2001*). RNAi cultures were grown overnight in 50 ml conical tubes at 37°C with shaking at 220 RPM in 5 ml LB medium containing 50 μg/ml carbenicillin and 12.5 μg/ml tetracycline. Cultures were diluted 1:5 the following day in LB containing carbenicillin and tetracycline to allow for re-entry into the logarithmic growth phase, grown to an $OD_{600}$ of 1.5 (~6 hr). Cultures were centrifuged at 4500 RPM for 10 min, concentrated to a volume of 5 ml, and then induced with 1 mM IPTG prior to plating. Bacterial cultures were seeded onto standard NGM plates containing 50 μg/ml carbenicillin, 12.5 μg/ml tetracycline, and 0.4 mM IPTG.

## Lifespans

Worms were synchronized by timed egg lay, upshifted to 25°C at the L2 stage, then scored for lifespan at 25°C or 20°C, as previously described (*Arantes-Oliveira et al., 2002*; *Robida-Stubbs et al., 2012*). For analyses of *glp-1(ts)* at 20°C, worms were downshifted from 25°C upon reaching adulthood. Animals were transferred at the first day of adulthood to fresh plates containing FUdR (ACROS Organics/Thermo Fisher Scientific, Geel, Belgium) at a concentration of 100 μg/ml to inhibit progeny development (*Mitchell et al., 1979*), unless otherwise indicated. RNAi-treated worms were placed on RNAi feeding plates starting at the first day of adulthood. Worms were maintained at a density of 30 worms per 6 cm plate on live bacteria and scored every other day. Animals that crawled off the plate, ruptured, or died from internal hatching were censored. Lifespans were graphed as Kaplan–Meier survival curves with JMP Pro 12 (SAS Institute, Middleton, MA). p values for survival curve analysis were generated using log-rank test. Additional statistical analysis was performed with GraphPad Prism 6 (GraphPad Software, La Jolla, CA). p values for mean lifespan analysis were calculated by two-way ANOVA with post hoc Holm–Šídák correction.

## Stress assays

Synchronized animals were incubated at 25°C during development then scored for survival hourly beginning at either days 1 or 3 of adulthood. Feeding RNAi was started at the L1 stage for day-1 stress assays or post-developmentally at day-1 adulthood for stress assays performed at day-3 adulthood. For the arsenite stress assay, worms were incubated in M9 buffer containing 5 mM AS (Riedel-de Haën, Seelze, Germany). For the tert-butyl hydroperoxide (TBHP) stress assay, worms were placed on solid NGM plates containing 15.4 mM TBHP (Sigma–Aldrich, St. Louis, MO). TBHP plates were freshly prepared on the day of the experiment. Survival assays were graphed as Kaplan–Meier survival curves with JMP Pro 12. p values for survival curve analysis were generated using log-rank test. Additional statistical analysis was performed with GraphPad Prism 6. p values for mean survival analysis were calculated by two-way ANOVA with post hoc Holm–Šídák correction.

## Microscopy

Animals were anaesthetized for 5 min in 0.06% tetramisole/M9 buffer, mounted on 2% agarose pads on glass slides under coverslips, and imaged with ZEN 2012 software on an Axio Imager.M2 microscope (Zeiss, Jena, Germany).

## GFP reporter scoring

Intestinal SKN-1::GFP nuclear localization and *gst-4p*::GFP::NLS expression were scored as 'high', 'medium', or 'low' as previously described (*An and Blackwell, 2003*; *Ewald et al., 2015*). 'High' denotes strong intensity in all intestinal nuclei; 'medium' indicates relatively lower intensity or distribution in approximately half of intestinal nuclei; 'low' denotes weak or no visible GFP intensity in intestinal nuclei. p values were calculated by two-sided $\chi^2$ test.

Intestinal DAF-16::GFP nuclear localization was scored as 'high', 'medium', or 'low' as previously described (*Henderson and Johnson, 2001*; *Berman and Kenyon, 2006*; *Curran and Ruvkun, 2007*). 'High' denotes more DAF-16::GFP observed in the nucleus compared to the cytoplasm;

**Table 4.** qRT-PCR primers used in this study

| Gene | Sequence | Annotation | Primer pair |
|---|---|---|---|
| gst-4 | K08F4.7 | Glutathione S-transferase | FWD: CCCATTTTACAAGTCGATGG<br>REV: CTTCCTCTGCAGTTTTTCCA |
| F20D6.11 | F20D6.11 | Flavin-adenine dinucleotide (FAD)-binding oxidoreductase | FWD: GGAAATTCTCGGTAGAATCGAA<br>REV: ACGATCACGAACTTCGAACA |
| nit-1 | ZK1058.6 | Nitrilase | FWD: AATCCTCCGACTATCCCTTG<br>REV: AGCGAATCGTTTCTTTTGTG |
| rpn-6.1 | F57B9.10 | 19S non-ATPase subunit | FWD: AATATTGGAAAAGCACCTGAAATGT<br>REV: TTTGATGTGGAAGTGAAGTCATTGT |
| lipl-3 | R11G11.14 | Lysosomal triglyceride lipase | FWD: ATGGGCAGGCAAATCCACCA<br>REV: AGTTGTTCTGCGCAATTATA |
| *cdc-42 | R07G3.1 | Housekeeping gene | FWD: CTGCTGGACAGGAAGATTACG<br>REV: CTCGGACATTCTCGAATGAAG |
| *Y45F10D.4 | Y45F10D.4 | Housekeeping gene | FWD: GTCGCTTCAAATCAGTTCAG<br>CREV: GTTCTTGTCAAGTGATCCGACA |

Select primer sequences were obtained from previous publications (**Robida-Stubbs et al., 2012**; **Vilchez et al., 2012**; **O'Rourke and Ruvkun, 2013**).

'medium' indicates animals with noticeable DAF-16::GFP in the nucleus but higher levels in the cytoplasm; 'low' denotes entirely cytoplasmic DAF-16::GFP.

SKN-1::GFP color isolation was performed to reduce gut granule autofluorescence using selective color matching against rgb(99,159,94) with a fuzziness setting of 125 and auto contrast in Adobe Photoshop CC 2014 (Adobe, San Jose, CA). DAF-16::GFP color isolation was similarly performed using selective color matching against rgb(0,255,111) with a fuzziness setting of 100.

## qRT-PCR

Samples were prepared from ~200 day-3 adult worms synchronized by timed egg lay. RNA was extracted using TRIzol (Thermo Fisher, Waltham, MA)-based phenol-chloroform extraction and purified with RNA Clean and Concentrator-5 spin columns (Zymo Research, Irvine, CA). RNA concentration and quality was assessed with a NanoDrop 1000 spectrophotometer (Thermo Fisher). cDNAs were prepared using SuperScript III First-Strand Synthesis SuperMix for qRT-PCR (Thermo Fisher). mRNA levels were quantified from biological triplicates and technical duplicates using SYBR Green (Thermo Fisher) fluorescence on a 384-well format Real-Time PCR 7900 (Applied Biosystems, Foster City, CA). After an initial denaturation step (95°C for 10 min), amplification was performed using 40 cycles of denaturation (95°C for 15 s) and annealing (60°C for 1 min). Samples were analyzed by the standard curve method, with normalization to the reference genes cdc-42 and Y45F10D.4 (**Hoogewijs et al., 2008**). p values were calculated by two-sided Student's t-test with post hoc Holm-Šídák correction in GraphPad Prism 6. The primers used in this study are provided in **Table 4**.

## RNA sequencing (RNA-seq)

Samples were prepared from ~5000 synchronized, L1 arrested day-1 adult animals cultured at 25°C. Worms were synchronized by sodium hypochlorite (bleach) treatment, as previously described (**Porta-de-la-Riva et al., 2012**). Bleach solution (9 ml ddH2O; 1 ml 1 N NaOH; 4 ml Clorox bleach) was freshly prepared before each experiment. Worms were bleached for 5 min, washed 5× in M9, and arrested at the L1 stage at 25°C in M9 containing 10 µg/ml cholesterol. Feeding RNAi was started at the L1 stage. This approach only partially reduces skn-1 function but allows analysis of larger samples than would be feasible with skn-1 mutants, which are sterile (**Bowerman et al., 1992**). Because these animals were not treated with FUdR, the WT adults contained an intact germline and eggs. As is explained in the 'Results' section, we therefore confined our analysis to genes that were overrepresented in glp-1(ts) animals, which lack eggs and most of the germline, and established a high-confidence cutoff for genes that were upregulated by GSC absence as opposed to simply being expressed specifically in somatic tissues. RNA was extracted using the same protocol for qRT-PCR samples. Purified RNA samples were DNase treated and assigned an RNA Integrity Number (RIN) quality score using a Bioanalyzer 2100 (Agilent Technologies, Santa Clara, CA). Only

matched samples with high RIN scores were sent for sequencing. Single read 50 bp RNA-seq with poly(A) enrichment was performed at the Dana-Farber Cancer Institute Center for Computational Biology using a HiSeq 2000 (Illumina, San Diego, CA).

FASTQ output files were aligned to the WBcel235 (February 2014) *C. elegans* reference genome using STAR (*Dobin et al., 2013*). These files have been deposited at the Gene Expression Omnibus (GEO) with the accession number GSE63075. Samples averaged 75% mapping of sequence reads to the reference genome. Differential expression analysis was performed using a custom R and Bioconductor RNA-seq pipeline (http://bioinf.wehi.edu.au/RNAseqCaseStudy/) (*Gentleman et al., 2004*; *Anders et al., 2013*; *R Core Team, 2014*). Quantification of mapped reads in the aligned SAM output files was performed using featureCounts, part of the Subread package (*Liao et al., 2013*, *2014*). We filtered out transcripts that didn't have at least one count per million reads in at least two samples. Quantile normalization and estimation of the mean–variance relationship of the log counts was performed by voom (*Law et al., 2014*). Linear model fitting, empirical Bayes analysis, and differential expression analysis were then conducted using limma (*Smyth, 2005*). To identify genes that are upregulated in a SKN-1-dependent manner by GSC loss, we sought genes for which *glp-1(ts)* expression was higher than WT, and for which *glp-1(ts);skn-1(−)* expression was reduced relative to *glp-1(ts)*. To test for this pattern, if a gene's expression change was higher in the comparison of *glp-1(ts)* vs WT and lower in the comparison of *glp-1(ts);skn-1(−)* vs *glp-1(ts)*, then we calculated the minimum (in absolute value) of the *t*-statistics from these two comparisons, and assessed the significance of this statistic by comparing to a null distribution derived by applying this procedure to randomly generated *t*-statistics. We corrected for multiple testing in this and the differential expression analysis using the false discovery rate (*Benjamini and Hochberg, 1995*). Heatmaps were generated using heatmap.2 in the gplots package (*Warnes et al., 2014*).

Functional annotations and phenotypes were obtained from WormBase build WS246. SKN-1 transcription factor binding site analysis of hits was conducted with biomaRt, GenomicFeatures, JASPAR, MotifDb, motifStack, MotIV, and Rsamtools (*Sandelin et al., 2004*; *Durinck et al., 2005*, *2009*; *Lawrence et al., 2013*; *Ou et al., 2013*; *Mercier and Gottardo, 2014*; *Shannon, 2014*). JASPAR analysis was performed with the SKN-1 matrix MA0547.1 using 2 kb upstream sequences obtained from Ensembl WBcel235 (*Staab et al., 2013a*). modENCODE SKN-1::GFP ChIP-seq analysis of hits was performed using biomaRt, ChIPpeakAnno, IRanges, and multtest (*Durinck et al., 2005*, *2009*; *Gerstein et al., 2010*; *Zhu et al., 2010*; *Niu et al., 2011*; *Lawrence et al., 2013*). SKN-1::GFP ChIP-seq peaks were generated by Michael Snyder's lab. We used the peak data generated from the first 3 larval stages: L1 (modENCODE_2622; GSE25810), L2 (modENCODE_3369), and L3 (modENCODE_3838; GSE48710). Human ortholog matching was performed using WormBase, Ensembl, and OrthoList (*Shaye and Greenwald, 2011*). Gene lists were evaluated for functional classification and statistical overrepresentation with Database for Annotation, Visualization, and Integrated Discovery (DAVID) version 6.7 (*Dennis et al., 2003*).

### *rpn-6.1* binding site analysis

SKN-1 and DAF-16 binding peaks within the first intron and the promoter of *rpn-6.1* were previously identified by the modENCODE project (*Furuyama et al., 2000*; *Gerstein et al., 2010*; *Niu et al., 2011*). We identified multiple hits with the consensus binding sequence ATCAT in the TRANSFAC matrices N\$SKN1_01 and N\$SKN1_02 using MATCH (Biobase, Wolfenbüttel, Germany) that overlap with the SKN-1::GFP ChIP-seq binding peaks within the first intron and the promoter of *rpn-6.1* (*Kel et al., 2003*; *Matys et al., 2006*). Our analysis also confirmed a previously identified hit (*Vilchez et al., 2012*) in the N\$DAF16_01 matrix with the consensus binding sequence TGTTT that overlaps with the DAF-16::GFP ChIP-seq peak within the first intron. No putative DAF-16 binding sites were identified in the *rpn-6.1* promoter in the TRANSFAC MATCH analysis.

### Proteasome activity

In vitro chymotrypsin-like proteasome activity assays were performed as previously described (*Kisselev and Goldberg, 2005*; *Vilchez et al., 2012*). Worms were bleach synchronized and maintained at 25°C from egg stage, then lysed at day 1 of adulthood, unless otherwise noted, in freshly prepared proteasome activity assay buffer (50 mM Tris–HCl, pH 7, 250 mM sucrose, 5 mM MgCl$_2$, 0.5 mM EDTA, 2 mM ATP, and 1 mM dithiothreitol) using a Branson digital sonifier at 4°C. Lysates

were centrifuged at 10,000×*g* for 15 min at 4°C. 25 μg of protein, calculated using the BCA protein assay (#23225; Pierce Biotechnology/Thermo Fisher, Rockford, IL), was transferred to a flat 96-well microtiter plate (Nunc, Roskilde, Denmark). Samples were incubated at 25°C, and fluorogenic chymotrypsin substrate (#230914; Calbiochem/EMD Millipore, San Diego, CA) was added to the plate immediately before analysis. Fluorescence (380 nm excitation; 460 nm emission) was measured every 3 min for 1 hr at 25°C using a Synergy MX microplate reader with Gen5 software (Bio-Tek, Winooski, VT). Lysates were assayed in triplicate. p values were calculated by two-sided Student's *t*-test in GraphPad Prism 6.

### Fixed ORO staining

ORO staining was performed on fixed animals, essentially as described (*O'Rourke et al., 2009*; *Yen et al., 2010*), with some modifications. 200–300 day-1 adult worms synchronized by timed egg lay were washed three times with phosphate-buffered saline (PBS) then snap frozen in a dry ice/ethanol bath. Upon thawing, worms were treated with PBS containing 2% paraformaldehyde (PFA) (#15713; Electron Microscopy Services, Hatfield, PA), using three freeze thaw cycles with dry ice/ethanol to permeabilize the cuticle. Worms were then washed with PBS to remove the PFA. Filtered ORO solution (0.5 g of ORO powder [#O0625; Sigma–Aldrich] in 100 ml of 60% isopropanol) was prepared freshly before each experiment. Worms were stained for 3 hr in a round bottom 96 well plate in ORO solution at room temperature with gentle shaking. Longer staining periods, such as overnight incubation (*O'Rourke et al., 2009*), saturated ORO staining in *glp-1(ts)* animals to a level that rendered *glp-1(ts)* and *glp-1(ts);skn-1* strains indistinguishable.

Animals were imaged at 40× using differential interference contrast microscopy. Quantification of ORO staining was performed on the upper intestine, directly below the pharynx. Since ORO absorbs light at 510 nm (green channel), we performed background subtraction of the red channel from the green channel in Adobe Photoshop CC (Adobe) to specifically isolate the ORO staining, as previously described (*Yen et al., 2010*). Quantification of mean intensity over background for each animal was performed using Fiji (http://fiji.sc). Statistical analysis was performed with GraphPad Prism 6. p values were calculated by one-way ANOVA with post hoc Holm-Šídák correction.

### DHS-3::GFP scoring

Using a COPAS Biosort (Union Biometrica, Holliston, MA) (*Pulak, 2006*), bleach-synchronized day-1 adult worms were scored for GFP fluorescence. RNAi was initiated after L1 arrest. The COPAS was used to record three attributes for each individual nematode: time of flight (TOF), which corresponds to nematode length; extinction (EXT), which corresponds to the optical density; and GFP fluorescence intensity. TOF and EXT measurements are related to the size and age of the nematode; both increase with development. These parameters were used to specifically gate adult worms. GFP fluorescence was normalized to worm size as a ratio of GFP/TOF values. Representative GFP images of each strain were captured at 4× using an Olympus IX51 inverted microscope (Olympus, New Orleans, LA). p values were calculated by one-way ANOVA with post-hoc Holm-Šídák correction in GraphPad Prism 6.

### Triglyceride quantification

Triglyceride (TAG) levels were measured with the Triglyceride Colorimetric Assay Kit (#10010303; Cayman Chemical, Ann Arbor, MI). Samples were run according to the manufacturer's protocol in triplicate. TAG concentrations were normalized relative to protein concentration using the BCA protein assay (Pierce Biotechnology).

## Acknowledgements

We thank Adam Antebi, Cynthia Kenyon, and Pingsheng Liu for strains, Javier Apfeld, William Mair, and Blackwell lab members for helpful discussions, and Sneha Rath and Collin Ewald for contributions to early stages of this work. Some strains were provided by the *Caenorhabditis* Genetics Center (CGC), which is funded by NIH Office of Research Infrastructure Programs (P40 OD010440).

## Additional information

### Funding

| Funder | Grant reference number | Author |
|---|---|---|
| National Institute of Diabetes and Digestive and Kidney Diseases (NIDDK) | NRSA Institutional Postdoctoral Training Grant, T32DK007260 | Michael J Steinbaugh |
| National Institute of General Medical Sciences (NIGMS) | R01GM062891 | T Keith Blackwell |
| National Institute of General Medical Sciences (NIGMS) | R01GM094398 | T Keith Blackwell |
| National Institute on Aging | R21AG043949 | T Keith Blackwell |
| Myra Reinhard Family Foundation | Research Award | T Keith Blackwell |
| National Institute of Diabetes and Digestive and Kidney Diseases (NIDDK) | Diabetes Research Center Award, P30DK036836 | T Keith Blackwell |
| National Institutes of Health (NIH) | Office of Research Infrastructure Programs (P40 OD010440) | T Keith Blackwell |

The funders had no role in study design, data collection and interpretation, or the decision to submit the work for publication.

### Competing interests
### Author contributions
MJS, Conception and design, Acquisition of data, Analysis and interpretation of data, Drafting or revising the article; SDN, Conception and design, Acquisition of data, Analysis and interpretation of data; SR-S, SR-S, Conception and design, Acquisition of data, Analysis and interpretation of data; PR, Conception and design, Acquisition of data, Analysis and interpretation of data; LEMM, Acquisition of data; JMH, Acquisition of data; TNO, Acquisition of data; RE, Acquisition of data; JMD, Analysis and interpretation of data; TKB, Conception and design, Analysis and interpretation of data, Drafting or revising the article

### Author ORCIDs
Michael J Steinbaugh, http://orcid.org/0000-0002-2403-2221

## Additional files

### Supplementary files
• Supplementary file 1. (a) List of genes activated by GSC absence. The gene list is sorted by functional grouping, then by fold change of mRNA expression in *glp-1(ts)* relative to wild type. A fold change cutoff of 4 (p < 0.05; n = 1306) was used (see *Figure 3—figure supplement 1* for rationale). A more conservative fold change cutoff of 5 that captured fewer genes (n = 615) was used for DAVID cluster analysis. FDR denotes false discovery rate. (b) List of genes activated by SKN-1 in wild-type animals. The gene list is sorted by functional grouping, then by fold change of mRNA expression in WT *skn-1* RNAi-treated worms relative to WT control vector-treated worms. A fold change cutoff of 0.67 (n = 295) was used to enrich for higher confidence SKN-1 targets. (c) List of genes activated by SKN-1 in GSC(−) animals. The gene list is sorted by functional grouping, then by fold change of mRNA expression in *glp-1(ts) skn-1* RNAi-treated worms relative to *glp-1(ts)* control vector-treated worms. A fold change cutoff of 0.67 (p < 0.05; n = 529) was used to enrich for higher confidence SKN-1 targets. (d) List of genes activated by GSC absence in a SKN-1-dependent manner. The gene list is sorted by functional grouping, then by fold change of mRNA expression in *glp-1 (ts) skn-1* RNAi-treated worms relative to *glp-1(ts)* control vector-treated worms. We employed a GSC(−) fold change cutoff of >4 and *skn-1* RNAi cutoff of <0.67 to generate the list (see *Figure 3—figure supplement 1* for GSC(−) FC cutoff rationale). Statistics were generated by min analysis.

Predicted SKN-1 binding sites were determined using 1.5 kb upstream sequences from WBcel235 and the SKN-1 JASPAR matrix (*Staab et al., 2013a*). SKN-1::GFP ChIP-seq binding analysis was performed using the L1, L2, and L3 data sets available from modENCODE (*Niu et al., 2011*). Additional details are available in 'Materials and methods'.

## Major datasets

The following dataset was generated:

| Author(s) | Year | Dataset title | Dataset URL | Database, license, and accessibility information |
|---|---|---|---|---|
| Steinbaugh MJ, Dreyfuss JM, Blackwell TK | 2015 | RNA-seq analysis of germline stem cell removal and loss of SKN-1 in C. elegans | http://www.ncbi.nlm.nih.gov/geo/query/acc.cgi?acc=GSE63075 | Publicly available at the NCBI Gene Expression Omnibus (Accession no: GSE63075). |

The following previously published datasets were used:

| Author(s) | Year | Dataset title | Dataset URL | Database, license, and accessibility information |
|---|---|---|---|---|
| Zhong M, Snyder M, Slightam C, Kim S, Murray J, Waterston R, Gerstein M, Niu W, Janette J, Raha D, Agarwal A, Reinke V, Sarov M, Hyman A | 2011 | Identification of Transcription Factor SKN-1::GFP Binding Regions in L1 | http://intermine.modencode.org/release-33/report.do?id=77000137 | Publicly available at modMine (Accession no: modENCODE_2622). |
| Zhong M, Snyder M, Slightam C, Kim S, Murray J, Waterston R, Gerstein M, Niu W, Janette J, Raha D, Agarwal A, Reinke V, Sarov M, Hyman A | 2013 | Identification of Transcription Factor SKN-1::GFP Binding Regions in L2 | http://intermine.modencode.org/release-33/report.do?id=77000379 | Publicly available at modMine (Accession no: modENCODE_3369). |
| Zhong M, Snyder M, Slightam C, Kim S, Murray J, Waterston R, Gerstein M, Niu W, Janette J, Raha D, Agarwal A, Reinke V, Sarov M, Hyman A | 2013 | Identification of Transcription Factor SKN-1::GFP Binding Regions in L3 | http://intermine.modencode.org/release-33/report.do?id=77000500 | Publicly available at modMine (Accession no: modENCODE_3838). |
| Zhong M, Snyder M, Slightam C, Kim S, Murray J, Waterston R, Gerstein M, Niu W, Janette J, Raha D, Agarwal A, Reinke V, Sarov M, Hyman A | 2013 | Identification of Transcription Factor SKN-1::GFP Binding Regions in L4 | http://intermine.modencode.org/release-33/report.do?id=77000600 | Publicly available at modMine (Accession no: modENCODE_4631). |
| Staab TA, Griffen TC, Corcoran C, Evgrafov O, Knowles JA, Sieburth D | 2013 | SKN-1 from the JASPAR CORE database | http://jaspar.genereg.net/cgi-bin/jaspar_db.pl?rm=browse&db=core&tax_group=nematodes | Publicly available at the JASPAR (Accession no: MA0547.1). |
| Zhong M, Snyder M, Slightam C, Kim S, Murray J, Waterston R, Gerstein M, Niu W, Janette J, Raha D, Agarwal A, Reinke V, Sarov M, Hyman A | 2011 | Identification of Transcription Factor SKN-1::GFP Binding Regions in L1 | http://www.ncbi.nlm.nih.gov/geo/query/acc.cgi?acc=GSE25810 | Publicly available at the NCBI Gene Expression Omnibus (Accession no: GSE25810). |

| Zhong M, Snyder M, Slightam C, Kim S, Murray J, Waterston R, Gerstein M, Niu W, Janette J, Raha D, Agarwal A, Reinke V, Sarov M, Hyman A | 2013 | Identification of Transcription Factor SKN-1::GFP Binding Regions in L3 | http://www.ncbi.nlm.nih.gov/geo/query/acc.cgi?acc=GSE48710 | Publicly available at the NCBI Gene Expression Omnibus (Accession no: GSE48710). |
|---|---|---|---|---|
| Zhong M, Snyder M, Slightam C, Kim S, Murray J, Waterston R, Gerstein M, Niu W, Janette J, Raha D, Agarwal A, Reinke V, Sarov M, Hyman A | 2013 | Snyder_SKN-1_GFP_L4 | http://www.ncbi.nlm.nih.gov/geo/query/acc.cgi?acc=GSE46772 | Publicly available at the NCBI Gene Expression Omnibus (Accession no: GSE46772). |

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
