## [Decision Letter]

Thank you for sending your work entitled “Lipid-mediated regulation of SKN-1/Nrf in response to germ cell ablation” for consideration at *eLif*e. Your article has been favorably evaluated by K VijayRaghavan (Senior editor), a Reviewing editor, and two reviewers.

The reviewers found this manuscript interesting and potentially suitable for publication in *eLife*. For instance, one reviewer wrote: “This is an excellent, well written, and data-rich manuscript. Along the way, the authors correct findings published in several high profile papers by others and provide a refreshing interpretation of the nature of the lipid accumulation in *glp-1* mutants and the role of SKN-1 in response to this lipid overload. As such, it is a dramatic advance for a field that has generally ignored the role of intestinal yolk and simply interpreted all lipids in the intestine as the equivalent of adipose-like storage depots. The authors also do a thorough job of incorporating a variety of mutants that have been previously implicated in *glp-1* lifespan extension in the context of their studies.”

Another reviewer wrote: “This study is very interesting because it explores the mechanism by which the conserved transcription factor NRF/SKN-1 acts to regulate longevity in response to deficiencies in the germline. This study also highlights an intriguing and novel germline-to-soma signaling for lifespan. Because SKN-1 is highly conserved throughout evolution and plays a role in metabolism in mammals, this study has important ramifications for age-related diseases in higher organisms. This study will appeal to a broad audience, including the fields of lipid metabolism, proteostasis, aging, and signaling.”

However both reviewers raised some questions. Please address these comments in a revised version.

1) The authors used “GSC ablation” and “*glp-1(ts)*” interchangeably throughout the text beginning with the Introduction. While *glp-1(ts)* worms are defective in GSC production, they still produce a partial germline, whereas worms that have undergone laser ablation of the germline would not produce GSCs at all. The authors should either tone done their terminology or for a few key assays, test worms that have undergone actual surgical ablation of the germline?

2) Figure 4: To make a statement regarding the role of *rpn-6.1* in regulating *glp-1(ts)* proteostasis or longevity in general, the authors should consider more functional tests. For example, is *rpn-6.1* required for *glp-1(ts)* longevity or for enhanced proteasome activity in *glp-1(ts)* worms.

3) Figure 6/D suggests that excessive accumulation of yolk fat may drive SKN-1 activations in the intestine. Would worms treated with *rme-2* RNAi, which presumably are high fat, exhibit high SKN-1 nuclear localization and beneficial phenotypes, such as stress resistance and longevity, associated with SKN-1 activation?

4) Figure 5—figure supplement 1: The DHS-3 gene encodes a fatty acid dehydrogenase/reductase that targets the mitochondria. The DHS-3 marker likely reveals levels of short chain fatty acid breakdown. Though the quantification data correlate with the ORO staining pattern in Figure 5/C very well, is DHS-3::GFP an accurate marker for lipid accumulation or lipid breakdown? In other words, are authors using lipid breakdown as a predictor of lipid accumulation in this case? Furthermore, are SKN-1 and SBP-1 regulators of *dhs-3* gene expression, where SKN-1 is a negative regulator and SBP-1 is positive regulator of *dhs-3*? If this is the case, one could expect a similar trend independently of lipid accumulation levels.

5) Are genes encoding vitellogenin proteins upregulated at the mRNA level in somatic tissues of GSC(−) worms? This might provide insight into whether high somatic VIT-2 levels are directly due to failure of vitellogenin import into oocytes or a transcriptional consequence of GCS loss.

6) While the overall point of the oil-red-o experiments is very convincing, there is a mismatch between the examples shown (Figure 5) and the corresponding quantifications (Figure 5). I recognize that this is really a problem of oil-red-o staining as it is not that suitable for quantification. One solution is to show a series of images for each condition in supplementary material so that the range of data can be seen. Alternatively (but not necessary), the authors can complement the studies by biochemical measurements of lipids.

---

## [Author Response]

*1) The authors used “GSC ablation” and “*glp-1(ts)*” interchangeably throughout the text beginning with the Introduction. While* glp-1(ts) *worms are defective in GSC production, they still produce a partial germline, whereas worms that have undergone laser ablation of the germline would not produce GSCs at all. The authors should either tone done their terminology or for a few key assays, test worms that have undergone actual surgical ablation of the germline*?

We didn’t intend to be misleading, but in hindsight agree that we should not have been so imprecise with terminology. While GSC ablation is a straightforward concept, *glp-1(ts)* animals raised at the non-permissive temperature exhibit a block in GSC proliferation, a dramatically reduced GSC number, and a failure of more mature germ cells to form. Unfortunately, the latter context is not so easily described by a simple term, and we are reluctant to use “*glp-1(ts)”* exclusively because it could make the manuscript less readable and accessible for readers who are not very familiar with *C. elegans*. To clarify things, we have modified the presentation of the work to explain in the Introduction how effects of GSC absence can be studied through either laser surgery, or genetic inhibition of GSC proliferation. We then explain that for simplicity we refer to either situation as “GSC(−) animals” or “GSC(−) longevity”, and state clearly at the beginning of the Results that in this study our analyses involved use of *glp-1(ts)* mutants. We have removed the terms “ablated” and “ablation” where the meaning could be ambiguous, and where necessary for sentence flow instead used terms like GSC “inhibition” or “absence”. I hope the editors will agree with us that this is a reasonable compromise that keeps the writing as simple as possible while avoiding being misleading.

*2)*Figure 4*: To make a statement regarding the role of* rpn-6.1 *in regulating* glp-1(ts) *proteostasis or longevity in general, the authors should consider more functional tests. For example, is* rpn-6.1 *required for* glp-1(ts) *longevity or for enhanced proteasome activity in* glp-1(ts) *worms*.

We apologize for any lack of clarity, but we did not intend to make claims concerning the role of *rpn-6.1* in *glp-1(ts)* or longevity in general. Those models were previously published in the [107] reference that we cited. With respect to *rpn-6.1*, we intended only to show only that its expression requires SKN-1, and that it is very likely to be a direct SKN-1 target. We have reorganized our presentation of our proteasome data accordingly, and have added evidence indicating that most if not all other proteasome subunit genes also appear to be activated by SKN-1 directly.

*3)*Figure 6*suggests that excessive accumulation of yolk fat may drive SKN-1 activations in the intestine. Would worms treated with* rme-2 *RNAi, which presumably are high fat, exhibit high SKN-1 nuclear localization and beneficial phenotypes, such as stress resistance and longevity, associated with SKN-1 activation*?

We agree that these are good experiments. We have expanded our *rme-2* studies so that we now show that *rme-2* RNAi increases SKN-1::GFP nuclear localization, *skn-1-*dependent *gst-4p::GFP* activation, and *skn-1*-dependent stress resistance (Figure 6 and Table 2). These observations fulfill a crucial prediction of our model that excessive yolk fat drives SKN-1 activation in GSC(−) animals.

*4)*Figure 5—figure supplement 1*: The DHS-3 gene encodes a fatty acid dehydrogenase/reductase that targets the mitochondria. The DHS-3 marker likely reveals levels of short chain fatty acid breakdown. Though the quantification data correlate with the ORO staining pattern in*Figure 5*very well, is DHS-3::GFP an accurate marker for lipid accumulation or lipid breakdown? In other words, are authors using lipid breakdown as a predictor of lipid accumulation in this case? Furthermore, are SKN-1 and SBP-1 regulators of* dhs-3 *gene expression, where SKN-1 is a negative regulator and SBP-1 is positive regulator of* dhs-3*? If this is the case, one could expect a similar trend independently of lipid accumulation levels*.

In a study published 15 years ago, a computational algorithm predicted with 68% accuracy that the putative fatty acid hydrogenase/reductase DHS-3 localizes to mitochondria (WormBase annotation). Since then, an extensive series of proteomic and in vivo analyses in *C. elegans* have demonstrated conclusively that DHS-3 is instead present almost exclusively on the surface of intestinal lipid droplets, and marks the vast majority of these lipid droplets. This evidence includes a comprehensive set of experiments that were published in the [114] paper that we referenced, as well as new studies from the same lab that have been published since our paper was initially submitted (73). Both papers are referenced in the revised manuscript. These studies have experimentally validated DHS-3 as a marker of intestinal lipid droplets in vivo. The idea that this lipid metabolism enzyme is present almost exclusively on lipid droplets is consistent with the growing body of evidence that lipid droplets are complex organelles that incorporate many lipid metabolism enzymes, and act as hubs for multiple steps in lipid metabolism. It is appropriate to use DHS-3 as an indirect marker of fat storage, but we have examined its levels only as a backup modality to our oil-red-O staining. In the revised paper we have made this clear, and provided additional background about DHS-3.

With respect to expression of the *dhs-3* mRNA, the critical issue for our conclusions is whether its levels are regulated by GSCs, and dependent upon *skn-1*. Like numerous lipid metabolism enzymes, DHS-3 might be controlled by *sbp-1*, but we had added *sbp-1* RNAi to our analysis simply as a control, because it would call our conclusions into question if DHS-3 levels were not reduced by *sbp-1* knockdown. Our RNA-Seq data reveal that the relative representation of *dhs-3* mRNA in GSC(−) animals corresponds to the expectation for a somatic gene that is not regulated by GSCs (see discussion of somatic-specific mRNAs in Figure 3—figure supplement 1). More importantly, we find that *dhs-3* expression is unaffected by *skn-1* RNAi in either WT or GSC(−) animals. This is now clearly shown and discussed (see Figure 5—figure supplement 3). These results are consistent with DHS-3 acting as an indicator of intestinal lipid accumulation, as is expected given its reliable presence in intestinal lipid droplets, rather than being upregulated by SKN-1 independently of lipid levels.

*5) Are genes encoding vitellogenin proteins upregulated at the mRNA level in somatic tissues of GSC(−) worms? This might provide insight into whether high somatic VIT-2 levels are directly due to failure of vitellogenin import into oocytes or a transcriptional consequence of GCS loss*.

Our RNA-Seq data indicate that the VIT genes are not transcriptionally upregulated in GSC(−) worms (Figure 5—figure supplement 3). Most are not present at increased levels at all, and certainly none of these mRNAs are preferentially represented in GSC(−) samples more than would be expected from any somatic-specific gene. This supports our model that the high somatic VIT-2 levels derive directly from failure of vitellogenin import.

*6) While the overall point of the oil-red-o experiments is very convincing, there is a mismatch between the examples shown (*Figure 5*) and the corresponding quantifications (*Figure 5*). I recognize that this is really a problem of oil-red-o staining as it is not that suitable for quantification. One solution is to show a series of images for each condition in supplementary material so that the range of data* can *be seen. Alternatively (but not necessary), the authors* can *complement the studies by biochemical measurements of lipids*.

This was a very good suggestion, and after looking more closely at our individual images we have adjusted our presentation of the ORO staining data in Figure 5 in order to more accurately reflect the middle ground for these images. Representative images with staining closer to the mean (graphed in Figure 5) are now shown. To aid the reader in evaluating our staining, we have also compared these images to additional ones for each strain in Figure 5—figure supplement 1. In this new supplemental figure, for each experimental condition five representative images that represent approximate quintiles of staining quantitation are ordered by lowest to highest staining. Next to each bright-field image is the background-subtracted image that shows the selected boundary of the upper intestine and quantification of that region (see ORO section in Methods for additional detail). Images are shown for *skn-1* mutant, *skn-1* RNAi, and *daf-16* mutant experiments. As an alternative method for assessing lipid accumulation, we consider DHS-3::GFP to be a reliable indirect indicator of intestinal lipid storage (see above), but we also confirmed that *skn-1* mutants have increased triglyceride levels using a colorimetric triglyceride assay kit (Figure 5—figure supplement 2).